# OAT-FM: Optimal Acceleration Transport for Improved Flow Matching

## Abstract

As a powerful technique in generative modeling, Flow Matching (FM) aims to learn velocity fields from noise to data, which is often explained and implemented as solving Optimal Transport (OT) problems. In this study, we bridge FM and the recent theory of Optimal Acceleration Transport (OAT), developing an improved FM method called OAT-FM and exploring its benefits in both theory and practice. In particular, we demonstrate that the straightening objective hidden in existing OT-based FM methods is mathematically equivalent to minimizing the physical action associated with acceleration defined by OAT. Accordingly, instead of enforcing constant velocity, OAT-FM optimizes the acceleration transport in the product space of sample and velocity, whose objective corresponds to a necessary and sufficient condition of flow straightness. An efficient algorithm is designed to achieve OAT-FM with low complexity. OAT-FM motivates a new two-phase FM paradigm: Given a generative model trained by an arbitrary FM method, whose velocity information has been relatively reliable, we can fine-tune and improve it via OAT-FM. This paradigm eliminates the risk of data distribution drift and the need to generate a large number of noise data pairs, which consistently improves model performance in various generative tasks.

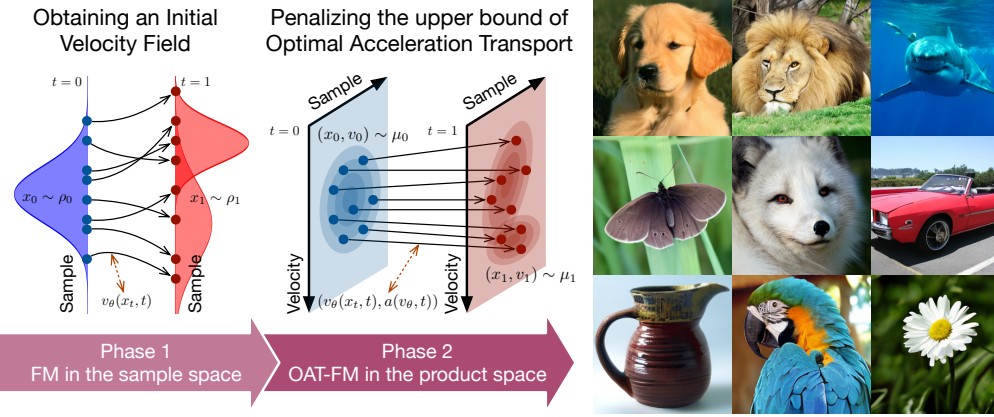

(a) A two-phase FM paradigm driven by OAT-FM   (b) Representative generation results

Figure 1: (a) The principle of OAT-FM and the corresponding two-phase FM paradigm. Given a pre-trained flow-based generator $v_\theta$, OAT-FM leverages its velocity field, fine-tuning the model by optimizing the flow in the product space of sample and velocity, which corresponds to an OAT problem. This flow captures the velocity $v_\theta$ and the acceleration $a$ determined by $v_\theta$ over time. (b) We train the state-of-the-art image generator SiT-XL (Ma et al., 2024) by our two-phase FM paradigm on the ImageNet $256 \times 256$ dataset (Deng et al., 2009), leading to high-quality image generation results. The results demonstrate that OAT-FM helps improve the cutting-edge model in practical high-dimensional generation tasks.

# 1 INTRODUCTION

As a promising generative modeling strategy, Flow Matching (FM) (Song et al., 2021a; Ho et al., 2020; Lipman et al., 2023) aims to learn a (deterministic or stochastic) velocity field capturing the transport of probability mass from a prior distribution (e.g., Gaussian noise) to a complex data distribution (e.g., natural images). In general, the learned velocity field corresponds to a neural solver of a specific ordinary or stochastic differential equation. Therefore, in the inference phase, we can simulate the differential equation with discrete sampling steps, resulting in the flows from noise to data. Nowadays, flow matching has achieved competitive performance in various high-dimensional data generative tasks, e.g., image generation (Lipman et al., 2023; Esser et al., 2024), audio generation (Liu et al., 2024; Wang et al., 2024), protein design (Bose et al., 2024; Yue et al., 2025), and so on.

Currently, some *two-phase FM paradigms* are proposed to achieve high-quality generation efficiently. Typically, given a well-trained flow/diffusion model, Rectified Flow (ReFlow) (Liu et al., 2023) and its variants (Lee et al., 2024; Hu et al., 2025) iteratively refit each learned flow trajectory to its linear or piecewise linear approximation. Such a so-called rectification phase straightens the flow progressively and thus allows a large sampling step size (and thus requires few steps) during inference. In addition, self-distillation methods, like Consistency Distillation (CD) (Song et al., 2023; Yang et al., 2024), treat the given model as a teacher and learn an efficient student model to fit the flow trajectories created by the teacher. Essentially, both these two-phase FM paradigms utilize the velocity field of the given model in their phase-2 training, enhancing model efficiency while preventing severe performance degradation.

In this study, we propose a novel improved FM method, called OAT-FM, based on the recent theory of Optimal Acceleration Transport (OAT) (Chen et al., 2018; Brigati et al., 2025), which provides a new way to leverage velocity information and leads to a new two-phase FM paradigm. As illustrated in Figure 1(a), OAT-FM minimizes the acceleration transport between the noise distribution and the data one defined in the product space of samples and their velocities. The implementation of OAT-FM corresponds to a bi-level optimization problem. In the lower-level, we solve an OAT problem, leveraging the endpoint velocity of flow to compute the Optimal Transport (OT) plan (or called coupling (Villani, 2021)) defined in the product space. In the context of FM, we can decompose the coupling and solve the OAT problem efficiently, whose complexity is the same with that of the classic OT problem (Peyré & Cuturi, 2019). Accordingly, we can sample noise-data pairs based on the OT plan during training. In the upper-level, we minimize an upper bound for the optimal acceleration transport from noise to data, which straightens flow trajectories with a theoretical guarantee. As shown in Figure 1(a), given an arbitrary flow/diffusion model, whose velocity information has been relatively reliable, we can continually train it via OAT-FM and improve its performance.

Different from existing OT-based FM methods (e.g., OT-CFM (Tong et al., 2024; Pooladian et al., 2023), OFM (Kornilov et al., 2024), and ReFlow (Liu, 2022; Hertrich et al., 2025)) and recent acceleration-driven FM methods (Chen et al., 2025a;b; Cao et al., 2025; Gong et al., 2025), OAT-FM applies a physically grounded objective tied directly to acceleration control and transport, whose straightening objective is mathematically equivalent to minimizing the physical action associated with acceleration defined by OAT. Compared with existing two-phase FM paradigms, such as ReFlow and CD, the paradigm based on OAT-FM neither requires generating a large number of paired training data nor relies on dense intermediate interpolation results, which eliminates the risk of data distribution drift. To our knowledge, OAT-FM makes the first attempt to explore the usefulness of acceleration-driven FM in high-dimensional generative tasks. As shown in Figure 1(b), the phase-2 training achieved by OAT-FM leads to promising high-resolution image generation results.

# 2 PROPOSED METHOD

## 2.1 OPTIMAL TRANSPORT-BASED FLOW MATCHING

Most existing FM methods fall into a generalized Conditional Flow Matching (CFM) framework (Tong et al., 2024). Denote $\mathbb{P}(\mathcal{X})$ as the set of probability measures defined in a sample space $\mathcal{X}$. Suppose that we have a data distribution $\rho_1 \in \mathbb{P}(\mathcal{X})$ and a noise one $\rho_0 \in \mathbb{P}(\mathcal{X})$, respectively. Typically, we set $\rho_0$ as a normal distribution $\mathcal{N}(0, 1)$. CFM models the evolutionary sample distribution from $\rho_0$ to $\rho_1$ conditioned on an auxiliary variable $z$, called conditional path and denoted

as $p_t(x|z)$ (with $t \in [0,1]$). It learns a neural network, denoted as $v_\theta$, to fit the velocity field $v_t(x|z)$ corresponding to $p_t(x|z)$, i.e.,

$$\min_\theta \quad \mathbb{E}_{z \sim \pi, \ t \sim \text{Unif}[0,1], \ x \sim p_t(\cdot|z)}[\|v_\theta(x,t) - v_t(x|z)\|^2], \tag{1}$$

where $\pi$ denotes the distribution of $z$. Once trained, the model generates new data from random noise by integrating parametrized velocities over time, i.e., $\hat{x}_1 = g_\theta(x_0) = x_0 + \int_0^1 v_\theta(x_t, t)\mathrm{d}t$, where $x_0 \sim \rho_0$. In practice, this generation process can be implemented by discrete Euler steps: Given the current $x_t$, we set a step size $\Delta t$, obtain $x_{t+\Delta t} = x_t + \Delta t \cdot v_\theta(x_t, t)$ and update the timestamp by $t \leftarrow t + \Delta t$. Repeating the above step till $t = 1$ leads to the generation result.

In the CFM framework, the distribution $\pi$ and the conditional path $p_t(x|z)$ play a central role, and different implementations result in various FM methods (Lim et al., 2024; Tong et al., 2024). For example, the early FM method in (Lipman et al., 2023) sets $\pi = \rho_1$ and $p_t(x|z)$ as a Gaussian distribution $\mathcal{N}(tz, (t\sigma - t + 1)^2)$. The independent coupling CFM (I-CFM) sets $\pi = \rho_0 \times \rho_1$, i.e., $z = (x_0, x_1)$ with $x_0 \sim \rho_0$ and $x_1 \sim \rho_1$ independently. Recently, some attempts have been made to interpret FM through the lens of optimal transport (Villani, 2021), leading to a series of optimal transport-based FM methods, e.g., OT-CFM (Tong et al., 2024; Pooladian et al., 2023) OFM (Kornilov et al., 2024), and kinetic FM (Shaul et al., 2023; 2025).

**Optimal Transport-based CFM (OT-CFM):** Denote the Wasserstein-2 distance between $\rho_0$ and $\rho_1$ as $\mathcal{W}_2(\rho_0, \rho_1)$. The dynamic (Benamou–Brenier) formulation of optimal transport (Benamou & Brenier, 2000) seeks a unique least–kinetic–energy flow $v$ corresponding to $\mathcal{W}_2^2(\rho_0, \rho_1)$, i.e.,

$$\mathcal{W}_2^2(\rho_0, \rho_1) = \min_{\rho, \, v} \int_0^1 \int_\mathcal{X} \frac{1}{2}\rho(x,t)\|v(x,t)\|_2^2 \mathrm{d}x\mathrm{d}t, \tag{2}$$

subject to the continuity equation $\partial_t \rho + \nabla_x \cdot (v\rho) = 0$ with boundary conditions $\rho(\cdot, 0) = \rho_0$ and $\rho(\cdot, 1) = \rho_1$. Accordingly, OT-CFM implements CFM by setting the distribution $\pi$ in (1) as the OT plan corresponding to $\mathcal{W}_2^2(\rho_0, \rho_1)$, leading to the following bi-level optimization problem:

$$\min_\theta \overbrace{\mathbb{E}_{(x_0, x_1) \sim \pi^*, \ t \sim \text{Unif}[0,1]}[\|v_\theta(x_t, t) - (x_1 - x_0)\|^2]}^{\text{Upper-level: } \mathcal{L}_\text{CFM}}, \quad \text{s.t. } \pi^* = \overbrace{\arg\min_{\pi \in \Pi(\rho_0, \rho_1)} \mathbb{E}_\pi[\|x_1 - x_0\|_2^2]}^{\text{Lower-level: } \mathcal{W}_2^2(\rho_0, \rho_1)}, \tag{3}$$

where $\Pi(\rho_0, \rho_1)$ denotes the set of couplings whose marginal distributions are $\rho_0$ and $\rho_1$, respectively. Note that we can set the conditional path as the deterministic linear interpolation between $x_0$ and $x_1$, i.e., given $z = (x_0, x_1)$, $p_t(x|z) = \delta_{x_t}$ with $x_t = (1-t) \cdot x_0 + t \cdot \hat{x}_1$. Accordingly, the velocity $v_t(x|z)$ becomes $x_1 - x_0$.

In theory, the objective function in (3) regresses $v_\theta(x_t, t)$ to the constant velocity $x_1 - x_0$. This constant velocity equals the characteristic velocity on the Wasserstein geodesic when $(x_0, x_1)$ are coupled optimally (McCann, 1997; Dong et al., 2024). In other words, training with $\pi^*$ aligns $v_\theta(x, t)$ with the least–kinetic–energy flow in (2), yielding straighter and more efficient flow trajectories. However, constant velocity is sufficient but not necessary for straightening flows.

**Proposition 1.** *The trajectory is straight if and only if the velocity direction is time invariant and the acceleration is everywhere parallel to the velocity. The classical (first-order) dynamical optimal transport is recovered as the special case with zero acceleration.*

Motivated by Proposition 1, we can move beyond first-order dynamics and minimize acceleration instead, which corresponds to the optimal acceleration transport problem in (Chen et al., 2018; Benamou et al., 2019; Brigati et al., 2025) and leads to the proposed OAT-FM method accordingly.

## 2.2 Flow Matching Based on Optimal Acceleration Transport

Given two distributions defined in the product space of sample and velocity, i.e., $\mu_0, \mu_1 \in \mathbb{P}(\mathcal{X} \times \mathcal{V})$, the optimal acceleration transport problem evolves a probability measure from $\mu_0$ to $\mu_1$ under deterministic second-order dynamics while minimizing total squared acceleration.[1]

---

[1]Obviously, the sample distribution $\rho_t$ is a marginal of $\mu_t$, i.e., $\rho_t(x) = \int_\mathcal{V} \mu_t(x, v)\mathrm{d}v$.

**Definition 1** (**Dynamic Formulation of Optimal Acceleration Transport (OAT)** (Benamou et al., 2019)). *Let $\mathcal{X} \subset \mathbb{R}^d$ be the sample space and $\mathcal{V} \subset \mathbb{R}^d$ the velocity space (by default $\mathcal{V} = \mathbb{R}^d$). For $\mu_0, \mu_1 \in \mathbb{P}(\mathcal{X} \times \mathcal{V})$, the optimal acceleration transport between them is defined as*

$$\mathcal{A}_2^2(\mu_0, \mu_1) := \min_{\mu, a} \int_0^1 \int_{\mathcal{X} \times \mathcal{V}} \frac{1}{2} \mu(x, v, t) \|a(x, v, t)\|_2^2 \, dx \, dv \, dt, \tag{4}$$

*subject to the Vlasov equation (Vlasov, 1968) $\partial_t \mu + v \cdot \nabla_x \mu + \nabla_v \cdot (a \mu) = 0$, with boundary conditions $\mu(\cdot, \cdot, 0) = \mu_0$ and $\mu(\cdot, \cdot, 1) = \mu_1$. Here, $a : \mathcal{X} \times \mathcal{V} \times [0, 1] \mapsto \mathbb{R}^d$ is the acceleration field, and the Vlasov equation expresses conservation of mass in the product space.*

Similar to the first-order optimal transport problem, OAT admits a static coupling problem on the product space in the Kantorovich format. [2]

**Definition 2** (**Kantorovich formulation of OAT** (Chen et al., 2018; Benamou et al., 2019; Brigati et al., 2025)). *Given $z_0 = (x_0, v_0) \sim \mu_0$ and $z_1 = (x_1, v_1) \sim \mu_1$, the OAT problem is equivalent to solving an optimal coupling w.r.t. squared acceleration cost, i.e.,*

$$\mathcal{A}_2^2(\mu_0, \mu_1) = \min_{\pi \in \Pi(\mu_0, \mu_1)} \mathbb{E}_{(z_0, z_1) \sim \pi} [c_\mathcal{A}^2(z_0, z_1)]$$

$$= \min_{\pi \in \Pi(\mu_0, \mu_1)} \mathbb{E}_{(z_0, z_1) \sim \pi} \Big[ 12 \underbrace{\Big\| \frac{x_1 - x_0}{T} - \frac{v_1 + v_0}{2} \Big\|^2}_{\text{velocity alignment}} + \underbrace{\|v_1 - v_0\|^2}_{\text{acceleration penalty}} \Big], \tag{5}$$

*where $T > 0$ denotes the time horizon defined between $\mu_0$ and $\mu_1$, which is 1 in our case.*

The OAT problem in (4) admits a minimizer $(\mu, a)$, while the OAT problem in (5) leads to an optimal coupling $\pi^* \in \Pi(\mu_0, \mu_1)$. Let $\Phi : (\mathcal{X} \times \mathcal{V})^2 \times [0, 1] \mapsto \mathcal{X} \times \mathcal{V}$ be an evaluation map associated with $a$. For $z_0, z_1 \sim \pi^*$, we have $\Phi_t(z_0, z_1) := (x_t, v_t)$. Then, for every time $t \in [0, 1]$, the distribution at time $t$ can be determined by the push-forward of $\pi^*$ through $\Phi_t$, i.e., $\mu_t = \Phi_t \# \pi^*$.

In the OAT problem, the optimal coupling is chosen by matching samples and velocities jointly in the product space, so sample alignment and velocity alignment are treated on the same footing. This means that points with similar velocity are more likely to be paired, and such alignments reduce the need for transverse corrections. Once velocities are aligned, minimizing the integrated acceleration emerges as the natural straightness objective. The following theorem indicates that OAT provides a second-order tool for straightening flow.

**Theorem 2** (**Straightening Flow via OAT**). *Given two boundary distributions $\mu_0, \mu_1 \in \mathbb{P}(\mathcal{X} \times \mathcal{V})$, OAT admits an optimal coupling $\pi^* \in \Pi(\mu_0, \mu_1)$ for the static problem in (5). For every $(x_0, v_0), (x_1, v_1) \sim \pi^*$, the corresponding trajectory is straight iff $v_0$ and $v_1$ are collinear with $x_1 - x_0$. Otherwise, it bends exactly to match the endpoints' orthogonal components.*

By adopting the OAT formulation, we shift from enforcing constant velocity to enforcing velocity alignment and acceleration minimization, which leads to the proposed OAT-FM method. **Essentially, OAT-FM is motivated by a desideratum: For a pre-trained flow model, whose velocity information is relatively reliable, refining it by solving an OAT problem can benefit its performance.** Suppose that we have derive a flow trajectory in $[0, 1]$ based on a model $v_\theta$, whose endpoints are $z_0 = (x_0, v_0)$ and $z_1 = (x_1, v_1)$, respectively. Given the model state at time $t$, denoted as $z_t(\theta) = (x_t, v_\theta(x_t, t))$, where $x_t = (1 - t) \cdot x_0 + t \cdot x_1$ and $t \in [0, 1]$, we define a cost function as follows

$$\ell_\mathcal{A}(z_0, z_1, t; \theta) = \alpha \Big\| \frac{x_t - x_0}{t} - \frac{v_0 + v_\theta(x_t, t)}{2} \Big\|_2^2 + (1 - \alpha) \|v_\theta(x_t, t) - v_0\|_2^2$$

$$+ \alpha \Big\| \frac{x_1 - x_t}{1 - t} - \frac{v_\theta(x_t, t) + v_1}{2} \Big\|_2^2 + (1 - \alpha) \|v_1 - v_\theta(x_t, t)\|_2^2 \tag{6}$$

$$= \frac{1}{13} \Big( c_\mathcal{A}^2(z_0, z_t(\theta)) + c_\mathcal{A}^2(z_t(\theta), z_1) \Big) \qquad \text{when } \alpha = \frac{12}{13}.$$

Obviously, this cost is based on the squared acceleration cost in (5), and we introduce a hyperparameter $\alpha \in [0, 1]$ to balance the term of velocity alignment and that of acceleration penalty. In practice, we can implement $\frac{x_t - x_0}{t}$ and $\frac{x_1 - x_t}{1 - t}$ equivalently by $x_1 - x_0$.

---

[2]To our knowledge, the Kantorovich formulation of OAT is first proposed and discussed by (Chen et al., 2018, Eq.10), and see also (Benamou et al., 2019, Eq.4.14) and (Brigati et al., 2025, Eq.8).

Given noise distribution $\mu_0$ and data distribution $\mu_1$, we can fine-tune the flow model by minimizing the expectation of $\ell_{\mathcal{A}}(z_0, z_1, t; \theta)$ over all $t \in [0, 1]$, $z_0 \sim \mu_0$, and $z_1 \sim \mu_1$, which corresponds to the following **OAT-FM problem**:

$$\min_{\theta} \overbrace{\mathbb{E}_{(z_0,z_1)\sim\pi^*, \, t\sim\text{Unif}[0,1]}[\ell_{\mathcal{A}}(z_0, z_1, t; \theta)]}^{\text{Upper-level: } \mathcal{L}_{\text{OAT}}(\mu_0,\mu_1;\alpha)}, \quad \text{s.t.} \ \ \pi^* = \overbrace{\underset{\pi\in\Pi(\mu_0,\mu_1)}{\arg\min} \ \mathbb{E}_{(z_0,z_1)\sim\pi}[c_{\mathcal{A}}^2(z_0, \, z_1)]}^{\text{Lower-level: } \mathcal{A}_2^2(\mu_0,\mu_1)}. \quad (7)$$

The learning problem in (7) considers an OAT-based flow-matching objective. In particular, the parameter $\alpha$ balances directional alignment with a proxy for small total acceleration, and the expectation over $t$ averages these effects along the path.

**Remark.** The connection between the OAT problem in (5) and our OAT-FM method is analogous to that between the OT problem and OT-CFM. In particular, it has been well known that OT-CFM learns a flow to achieve an optimal transport in the sample space, straightening flow trajectories by pursuing constant velocity (Tong et al., 2024). Our OAT-FM learns a flow to achieve optimal acceleration transport in the product space of sample and velocity, straightening flow trajectories by minimizing acceleration (i.e., a smooth velocity field).

The objective of OAT-FM provides a tight bound on the true OAT second-order discrepancy, ensuring effective minimization of the acceleration.

**Theorem 3** (**OAT Bound of OAT-FM**). *The OAT-FM objective $\mathcal{L}_{OAT}(\mu_0, \mu_1; \alpha)$ is lower-bounded by a scaled version of the true OAT second-order discrepancy, i.e.,*

$$\mathcal{L}_{OAT}(\mu_0, \mu_1; \alpha) \geq \frac{2}{27} \mathcal{A}_2^2(\mu_0, \mu_1), \quad (8)$$

*with $\alpha = 2/3$, and the equality held if and only if $v_1 = v_0$ for $\pi^*$-almost every pair.*

The proofs of all theorems can be found in Appendix B.

### 2.3 EFFICIENT IMPLEMENTATION

Similar to OT-CFM, the OAT-FM problem in (7) is a bi-level optimization problem as well. The lower-level problem determines a coupling $\pi^*$. The upper-level problem updates $\theta$ given $\pi^*$. In practice, we solve them via alternating optimization. Following the existing methods in (Pooladian et al., 2023; Tong et al., 2024), we solve the lower-level problem via min-batch approximation.

It should be noted that, although the coupling of OAT problem has a four-dimensional coupling, i.e., $\pi(z_0, z_1) = \pi(x_0, x_1, v_0, v_1)$, it has a decomposable structure in the context of FM. In particular, given an arbitrary sample $x$ and a time stamp $t$, we can determine its velocity as $v_\theta(x, t)$, which is conditionally independent of other samples or velocities. Therefore, we have $\pi(x_0, x_1, v_0, v_1) = \pi_x(x_0, x_1)\pi(v_0, v_1|x_0, x_1) = \pi_x(x_0, x_1)\pi_v(v_0|x_0)\pi_v(v_1|x_1)$, where $\pi_x$ is the marginal coupling associated with sample pairs and $\pi_v(\cdot|x_t) = \delta_{v_\theta(x,t)}$. As a result, we simplify the lower-level OAT problem in (7) as

$$\underset{\pi\in\Pi(\mu_0,\mu_1)}{\arg\min} \ \mathbb{E}_{(z_0,z_1)\sim\pi}\Big[c_{\mathcal{A}}^2(z_0, z_1)\Big] \Rightarrow \underset{\pi_x\in\Pi(\rho_0,\rho_1)}{\arg\min} \ \mathbb{E}_{(x_0,x_1)\sim\pi_x}\Big[12\|x_1-x_0-\bar{v}_{x_0,x_1}\|^2+\|\tilde{v}_{x_0,x_1}\|_2^2\Big], \quad (9)$$

where $\rho_0, \rho_1 \in \mathbb{P}(\mathcal{X})$ denote the noise and data distributions, $\bar{v}_{x_0,x_1} = \frac{1}{2}\big(v_\theta(x_0, 0) + v_\theta(x_1, 1)\big)$, and $\tilde{v}_{x_0,x_1} = v_\theta(x_1, 1) - v_\theta(x_0, 0)$. The reformulated problem in (9) becomes a classic OT problem (Peyré & Cuturi, 2019). As a result, the complexity of OAT-FM is the same as OT-CFM (Tong et al., 2024). The detailed derivation of (9) and the scheme of our learning algorithm are provided in Appendix C.

## 3 COMPARISONS WITH RELATED WORK

Different from existing OT-based flow matching methods (Tong et al., 2024; Pooladian et al., 2023; Kornilov et al., 2024), OAT-FM considers the coupling in the "sample-velocity" product space that minimizes the expected acceleration of flow, which fully leverages the endpoint velocity of flow to compute the OT plan and sampling noise-data pairs. Alongside OAT-FM, several works have

Table 1: A comparison of various FM methods based on first- and second-order dynamics

| Method | Dynamics | Parameterization | Training Loss | Space |
|---|---|---|---|---|
| OT-CFM | $\partial_t \rho + \nabla_x (v\rho) = 0$ | $v_\theta(x,t)$ | $\mathcal{L}_{\text{CFM}}$ in (3) | $\mathcal{X}$ |
| NRFlow / HOMO / SOM | $\partial_t^2 \rho + \nabla_x \cdot (v\,\partial_t \rho + a\rho) = 0$ | $v_{\theta_1}(x,t),\, a_{\theta_2}(x,t)$ | (11) / (12) / (13) | $\mathcal{X}$ |
| **OAT-FM** | $\partial_t \mu + \nabla_x (v\mu) + \nabla_v(a\mu) = 0$ | $v_\theta(x,t)$ | $\mathcal{L}_{\text{OAT}}$ in (7) | $\mathcal{X} \times \mathcal{V}$ |

extended diffusion/flow models based on second-order dynamics. Let $\rho_t \in \mathbb{P}(\mathcal{X})$. For $x_0 \sim \rho_0$ and $x_1 \sim \rho_1$, the rectified interpolation (Gong et al., 2025, Definition 3.12) between them, including the trajectory, the velocity, and the acceleration, can be written as $x_t = \alpha_t x_0 + \beta_t x_1$, $v_t = \dot{\alpha}_t x_0 + \dot{\beta}_t x_1$, and $a_t = \ddot{\alpha}_t x_0 + \ddot{\beta}_t x_1$, where $\{\alpha_t, \beta_t, \dot{\alpha}_t, \dot{\beta}_t, \ddot{\alpha}_t, \ddot{\beta}_t\}$ are predefined time–dependent coefficients. When $\rho_t$ satisfies the continuity equation $\partial_t \rho_t(x) + \nabla_x \cdot \big(v_t(x), \rho_t(x)\big) = 0$, differentiating it once more in time yields a second–order continuity law (Gong et al., 2025, Lemma 5.1):

$$\partial_t^2 \rho_t + \nabla_x \cdot (v_t \partial_t \rho_t + a_t \rho_t) = 0, \tag{10}$$

which couples density acceleration to both the velocity and acceleration fields.

Building on this perspective, recent second–order flow matching methods incorporate acceleration into their training losses. NRFlow (Chen et al., 2025b) augments the standard OT-CFM objectives by regressing a neural velocity predictor $v_{\theta_1}(x,t)$ to the rectified velocity $v_t$, and simultaneously regressing an acceleration predictor $a_{\theta_2}(v,x,t)$ to the rectified acceleration $a_t$. The resulting loss is

$$\mathcal{L}_{\text{NRFlow}} = \mathbb{E}_{(x_0,x_1)\sim\pi,\, t\sim\text{Unif}[0,1]}[\|v_t - v_{\theta_1}(x_t,t)\|_2^2 + \|a_t - a_{\theta_2}(v_{\theta_1}(x_t,t), x_t, t)\|_2^2]. \tag{11}$$

HOMO (Chen et al., 2025a) extends NRFlow by adding a self-consistency penalty, ensuring that the instantaneous velocity matches the average of the velocity and its one-step update $x_{t+d}$, i.e.,

$$\mathcal{L}_{\text{HOMO}} = \mathcal{L}_{\text{NRFlow}} + \mathbb{E}_{(x_0,x_1)\sim\pi,\, t\sim\text{Unif}[0,1]}[\|v_{\theta_1}(x_t,t) - \bar{v}_t\|_2^2], \tag{12}$$

with the target velocity defined as $\bar{v}_t = \frac{1}{2}(v_{\theta_1}(x_t,t) + v_{\theta_1}(x_{t+d}, t+d))$. Inspired by Mean-Flow (Geng et al., 2025), SOM (Cao et al., 2025) replaces pointwise supervision with interval averages and aligns the model to averaged quantities over $[r,t]$, whose the training loss is

$$\mathcal{L}_{\text{SOM}} = \mathbb{E}_{(x_0,x_1)\sim\pi,\, r\sim\text{Unif}[0,1],\, t\sim\text{Unif}[r,1]}[\|v_{\theta_1}(x_t,t) - \bar{v}_r(x_t)\|_2^2 + \|a_{\theta_2}(x_t,t) - \bar{a}_r(x_t)\|_2^2]. \tag{13}$$

where $\bar{v}(z_t, r, t) = \frac{1}{t-r}\int_r^t v(z_\tau, \tau)\,d\tau$ and $\bar{a}(z_t, r, t) = \frac{1}{t-r}\int_r^t a(z_\tau, \tau)\,d\tau$. Unfortunately, till now, the performance of these second-order FM methods in high-dimensional data generation tasks (e.g., high-resolution image generation) has not been verified.

Different from the above methods, OAT-FM introduces a physically grounded objective tied directly to acceleration control and second-order transport. As shown in Table 1, the key distinction lies in the underlying dynamics. OAT-FM evolves distributions in the joint space of sample and velocity according to Vlasov equation $\partial_t \mu(x,v) + \nabla_x \cdot (v\mu) + \nabla_v \cdot (a\mu) = 0$. This law enforces at the population level the intimate coupling between the density of states and the acceleration field that drives them. In contrast, NRFlow, HOMO, and SOM abide the classical continuity law and its time derivative (10), and train by regressing point-wise velocity and acceleration signals along a prescribed rectified path. Such formulations provide local supervision but do not impose a closed second-order transport constraint on the distribution as a whole. By casting training in the Vlasov flow rationale, OAT-FM thus $i$) encodes second-order conservation directly during training, $ii$) regularizes acceleration rather than learning additional acceleration models, and $iii$) avoids dependence on a specific time parametrization or rectification rule for supervision.

## 4 EXPERIMENTS

We evaluate the efficacy of OAT-FM in various tasks, from low-dimensional optimal transport to high-dimensional image generation. This section shows representative experimental results. More experimental results and implementation details are included in Appendix D.

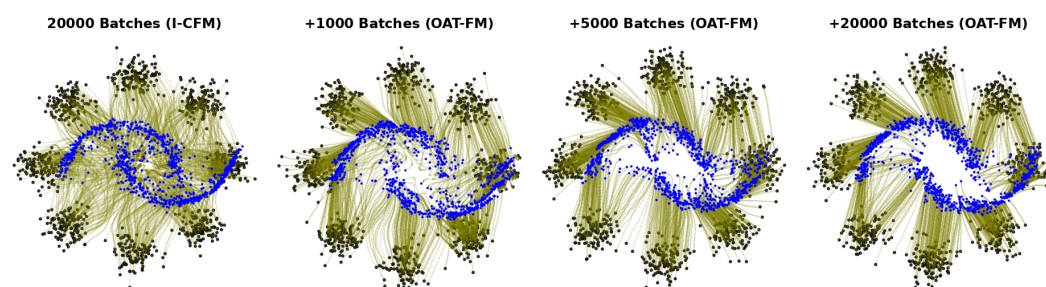

| 20000 Batches (I-CFM) | +1000 Batches (OAT-FM) | +5000 Batches (OAT-FM) | +20000 Batches (OAT-FM) |

Figure 2: An illustration of refining the flow of I-CFM via OAT-FM on the eight Gaussians to the Moons dataset. We conduct this experiment based on the code base provided in (Tong et al., 2024).

Table 2: A comparison of various methods in terms of data fitting (2-Wasserstein) and optimal transport approximation (normalized path energy). We run each task in five trials and record the average performance and standard deviation.

| Task | $\mathcal{N} \to$8gs | | 8gs$\to$moons | | $\mathcal{N} \to$moons | | $\mathcal{N} \to$scurve | | moons$\to$8gs | |
|---|---|---|---|---|---|---|---|---|---|---|
| Method | $\mathcal{W}_2^2 \downarrow$ | NPE$\downarrow$ | $\mathcal{W}_2^2 \downarrow$ | NPE$\downarrow$ | $\mathcal{W}_2^2 \downarrow$ | NPE$\downarrow$ | $\mathcal{W}_2^2 \downarrow$ | NPE$\downarrow$ | $\mathcal{W}_2^2 \downarrow$ | NPE$\downarrow$ |
| FM | $0.58_{\pm0.16}$ | $0.24_{\pm0.01}$ | $5.80_{\pm0.06}$ | $0.05_{\pm0.02}$ | $0.15_{\pm0.07}$ | $0.27_{\pm0.05}$ | $\mathbf{0.81}_{\pm0.39}$ | $0.08_{\pm0.04}$ | $7.39_{\pm0.45}$ | $0.96_{\pm0.05}$ |
| +OAT-FM | $\mathbf{0.31}_{\pm0.09}$ | $\mathbf{0.02}_{\pm0.01}$ | $\mathbf{0.08}_{\pm0.03}$ | $\mathbf{0.01}_{\pm0.01}$ | $\mathbf{0.08}_{\pm0.03}$ | $\mathbf{0.03}_{\pm0.01}$ | $0.90_{\pm0.18}$ | $\mathbf{0.03}_{\pm0.02}$ | $\mathbf{0.28}_{\pm0.10}$ | $\mathbf{0.04}_{\pm0.02}$ |
| I-CFM | $0.45_{\pm0.18}$ | $0.30_{\pm0.01}$ | $0.18_{\pm0.05}$ | $1.40_{\pm0.05}$ | $0.11_{\pm0.03}$ | $0.52_{\pm0.06}$ | $1.16_{\pm0.47}$ | $0.03_{\pm0.03}$ | $0.74_{\pm0.12}$ | $1.19_{\pm0.06}$ |
| +OAT-FM | $\mathbf{0.32}_{\pm0.10}$ | $\mathbf{0.04}_{\pm0.01}$ | $\mathbf{0.15}_{\pm0.03}$ | $\mathbf{0.13}_{\pm0.01}$ | $\mathbf{0.07}_{\pm0.02}$ | $\mathbf{0.04}_{\pm0.04}$ | $\mathbf{1.12}_{\pm0.45}$ | $\mathbf{0.03}_{\pm0.02}$ | $\mathbf{0.50}_{\pm0.11}$ | $\mathbf{0.44}_{\pm0.03}$ |
| VP-CFM | $0.43_{\pm0.14}$ | $0.24_{\pm0.01}$ | $0.15_{\pm0.02}$ | $1.24_{\pm0.05}$ | $0.10_{\pm0.03}$ | $0.31_{\pm0.07}$ | $\mathbf{1.05}_{\pm0.41}$ | $0.22_{\pm0.04}$ | $1.39_{\pm0.35}$ | $1.22_{\pm0.05}$ |
| +OAT-FM | $\mathbf{0.31}_{\pm0.12}$ | $\mathbf{0.03}_{\pm0.01}$ | $\mathbf{0.09}_{\pm0.01}$ | $\mathbf{0.02}_{\pm0.01}$ | $\mathbf{0.07}_{\pm0.02}$ | $\mathbf{0.04}_{\pm0.01}$ | $1.10_{\pm0.34}$ | $\mathbf{0.03}_{\pm0.02}$ | $\mathbf{0.32}_{\pm0.10}$ | $\mathbf{0.10}_{\pm0.02}$ |
| SB-CFM | $0.51_{\pm0.10}$ | $\mathbf{0.01}_{\pm0.01}$ | $0.13_{\pm0.04}$ | $0.03_{\pm0.01}$ | $\mathbf{0.08}_{\pm0.03}$ | $\mathbf{0.04}_{\pm0.03}$ | $\mathbf{0.79}_{\pm0.29}$ | $0.04_{\pm0.02}$ | $0.36_{\pm0.14}$ | $\mathbf{0.03}_{\pm0.02}$ |
| +OAT-FM | $\mathbf{0.34}_{\pm0.08}$ | $0.03_{\pm0.01}$ | $\mathbf{0.07}_{\pm0.01}$ | $\mathbf{0.01}_{\pm0.01}$ | $0.09_{\pm0.04}$ | $0.10_{\pm0.04}$ | $0.80_{\pm0.18}$ | $\mathbf{0.02}_{\pm0.02}$ | $\mathbf{0.25}_{\pm0.08}$ | $\mathbf{0.03}_{\pm0.02}$ |
| OT-CFM | $0.35_{\pm0.09}$ | $\mathbf{0.01}_{\pm0.01}$ | $\mathbf{0.07}_{\pm0.02}$ | $\mathbf{0.01}_{\pm0.01}$ | $0.07_{\pm0.02}$ | $\mathbf{0.04}_{\pm0.02}$ | $0.87_{\pm0.33}$ | $\mathbf{0.03}_{\pm0.03}$ | $0.31_{\pm0.10}$ | $\mathbf{0.02}_{\pm0.02}$ |
| +OAT-FM | $\mathbf{0.32}_{\pm0.10}$ | $0.04_{\pm0.01}$ | $\mathbf{0.07}_{\pm0.01}$ | $\mathbf{0.01}_{\pm0.01}$ | $\mathbf{0.06}_{\pm0.01}$ | $\mathbf{0.04}_{\pm0.01}$ | $\mathbf{0.83}_{\pm0.34}$ | $0.04_{\pm0.02}$ | $\mathbf{0.29}_{\pm0.09}$ | $0.10_{\pm0.02}$ |

### 4.1 TESTING ON THE LOW-DIMENSIONAL OT BENCHMARK

We first validate OAT-FM on the low-dimensional OT benchmark (Tong et al., 2024), which includes five 2D point cloud transport tasks. After applying FM (Lipman et al., 2023), I/SB/OT-CFM (Tong et al., 2024), and VP-CFM (Albergo et al., 2023), respectively, to train an MLP-based generator by 20,000 batches, we leverage OAT-FM to continually refine the model with the same number of batches. For fairness, we compare the models achieved by the two-phase FM paradigm with those trained by the baseline methods (i.e., FM, I/SB/OT-CFM, and VP-CFM) with 40,000 batches on two metrics: $i$) the 2-Wasserstein distance $\mathcal{W}_2^2(\hat{\rho}_1, \rho_1)$ between the distribution of generated samples $\hat{\rho}_1$ and that of target data $\rho_1$, and $ii$) the *Normalized Path Energy (NPE)* defined in terms of the 2-Wasserstein distance as

$$\text{NPE}(v_\theta) = \frac{|\text{PE}(v_\theta) - \mathcal{W}_2^2(\rho_0, \rho_1)|}{\mathcal{W}_2^2(\rho_0, \rho_1)}, \text{ where } \text{PE}(v_\theta) = \mathbb{E}_{x_0 \sim \rho_0} \int_0^1 \|v_\theta(x_t, t)\|^2 \, \mathrm{d}t. \quad (14)$$

Here, we generate all samples using the RK45 ODE solver (Dormand & Prince, 1980) with 101 integration steps from $t = 0$ to $t = 1$ and compute PE accordingly. This metric evaluates the transport cost of the learned flow relative to the dynamic optimal transport.

The quantitative results are presented in Table 2. We can find that with the help of OAT-FM, our phase-2 FM paradigm leads to better results in most situations. In particular, for those non-OT methods, e.g., FM, I-CFM, and VP-CFM, applying OAT-FM to achieve a second-phase training makes their flows fit dynamic optimal transport better, reducing the Wasserstein distance and NPE of each transport task significantly. For SB-CFM and OT-CFM, whose flows have been learned to fit dynamic optimal transports, applying OAT-FM can still make their models fit data distributions with lower Wasserstein distances while maintaining comparable NPE in general. Figure 2 visualizes the progressive straightening of I-CFM's transport trajectories on the eight Gaussian distributions (denoted as "8gs") to the Moons dataset during the OAT-FM refining process.

## 4.2 Unconditional Image Generation

Beyond the above low-dimensional OT benchmark, we further evaluate OAT-FM on generating CIFAR-10 images (Krizhevsky, 2009). We initialize our training from the pre-trained FM (Lipman et al., 2023), I-CFM (Tong et al., 2024), OT-CFM (Tong et al., 2024) and EDM (Karras et al., 2022) models, which serve as a starting point by providing good estimates for the boundary velocities. For FM, I-CFM, and OT-CFM, we follow the settings in (Tong et al., 2024). For EDM, same as (Lee et al., 2024), we adapt it into a flow matching model by adjusting its time and scaling factors, which allows a seamless transition to our phase-2 training (see Appendix D.1). All models are trained with a per-GPU batch size of 128 and an EMA decay rate of 0.9999. In the inference phase, we follow the standard setting, employing the Dopri5 solver for FM, I-CFM, and OT-CFM, and the Heun solver for EDM, respectively. For each model, we evaluate image quality using the Fréchet Inception Distance (FID) (Heusel et al., 2017) and record its number of training batches and that of inference steps (denoted as NFE) as well.

As shown in Table 3, OAT-FM consistently enhances the generation quality across all pre-trained models. Notably, for FM, I-CFM, and OT-CFM, our method achieves superior FID scores while requiring only 1K additional training batches, a substantial reduction from the 400K batches used by the original models. Furthermore, OAT-FM also improves upon the strong EDM baseline, lowering the FID from 1.96 to 1.93 with only 12K additional training batches. This result is also better than other competitive generative modeling methods, including the strong two-phase FM method 2-ReFlow++ (Lee et al., 2024). These results underscore that OAT-FM serves as an effective and computationally efficient plug-in module for refining existing unconditional generative models, thereby boosting their performance with minimal training overhead.

**Ablation Studies.** As shown in Table 4, we conduct ablation studies on CIFAR-10 to validate the design of OAT-FM. In particular, given the models trained by FM (Lipman et al., 2023) and EDM (Karras et al., 2022), respectively, we continually train them by OAT-FM under different settings, dissecting the contributions of its key components, i.e., OAT-based coupling computation in its lower-level problem and the OAT-based objective used in its upper-level problem. To adapt EDM for flow matching, we reparameterize its denoising as a velocity field predictor. This allows us to initialize our model with a strong, pre-trained diffusion backbone, facilitating a seamless transition to the second-phase OAT-FM training paradigm. The implementation details are in Appendix D.1.

Table 3: Comparisons of various methods in unconditional CIFAR-10 image generation. In the column "#Batch", the number of training batches of each baseline method is in black, while that of OAT-FM is in purple. The unit "K" means 1,000 batches. The results of the methods labeled by "∗" are from Lee et al. (2024).

| Method | #Batch | NFE↓ | FID↓ |
|---|---|---|---|
| FM (Lipman et al., 2023) | 400K | 147 | 3.71 |
| FM + OAT-FM | +1K | 135 | **3.54** |
| I-CFM (Tong et al., 2024) | 400K | 149 | 3.67 |
| I-CFM + OAT-FM | +1K | 138 | **3.48** |
| OT-CFM (Tong et al., 2024) | 400K | 132 | 3.64 |
| OT-CFM + OAT-FM | +1K | 126 | **3.46** |
| DDPM∗ (Ho et al., 2020) | | 1,000 | 3.17 |
| Score SDE∗ (Song et al., 2021b) | | 2,000 | 2.38 |
| LSGM∗ (Vahdat et al., 2021) | | 147 | 2.10 |
| 2-ReFlow++∗ (Lee et al., 2024) | | 35 | 2.30 |
| EDM (Karras et al., 2022) | | 35 | 1.96 |
| EDM + OAT-FM | +12K | 35 | **1.93** |

Table 4: Ablation studies of OAT-FM on CIFAR-10 dataset. The FID scores of the models trained under different settings are provided.

| Lower-level Problem | Upper-level Problem | Phase-1 Method FM | EDM |
|---|---|---|---|
| Without Phase-2 Training | | 3.71 | 1.96 |
| $\mathcal{W}_2^2$ in (3) | $\mathcal{L}_{\text{CFM}}$ in (3) | 3.75 | 8.77 |
| $\mathcal{W}_2^2$ in (3) | $\mathcal{L}_{\text{OAT}}$ in (9) | 3.55 | 8.68 |
| $\mathcal{A}_2^2$ in (9) | $\mathcal{L}_{\text{CFM}}$ in (3) | 3.81 | 1.95 |
| $\mathcal{A}_2^2$ in (9) | $\mathcal{L}_{\text{OAT}}$ in (9) | **3.54** | **1.93** |

When deriving the coupling by computing the Wasserstein distance in (3), the FM-based model maintains its low FID score on CIFAR-10 while EDM suffers severe performance degradation no matter what upper-level objective is. For the upper-level objective, we can find that $\mathcal{L}_{\text{OAT}}$ works better than $\mathcal{L}_{\text{CFM}}$ consistently for both FM- and EDM-based models. These results demonstrate that each component of OAT-FM plays an indispensable role in refining flows and enhancing model performance.

Table 5: A comparison on class-conditional image generation. In the column "#Epochs", the number of training epochs of each baseline method is in black, while that of OAT-FM is in purple.

| Method | #Epochs | FID↓ | sFID↓ | IS↑ | P↑ | R↑ |
|---|---|---|---|---|---|---|
| BigGAN-deep (Brock et al., 2019) | | 6.95 | 7.36 | 171.4 | 0.87 | 0.28 |
| StyleGAN-XL (Sauer et al., 2022) | | 2.30 | 4.02 | 265.1 | 0.78 | 0.53 |
| Mask-GIT (Chang et al., 2022) | | 6.18 | - | 182.1 | - | - |
| ADM-G/U (Dhariwal & Nichol, 2021) | | 3.94 | 6.14 | 215.8 | 0.83 | 0.53 |
| CDM (Ho et al., 2022) | | 4.88 | - | 158.7 | - | - |
| RIN (Jabri et al., 2023) | | 3.42 | - | 182.0 | - | - |
| Simple Diffusion$_{U-ViT, L}$ (Hoogeboom et al., 2023) | | 2.77 | - | 211.8 | - | - |
| VDM++ (Kingma & Gao, 2023) | | 2.12 | - | 267.7 | - | - |
| DiT-XL$_{CFG=1.5}$ (Peebles & Xie, 2023) | | 2.27 | 4.60 | 278.2 | 0.83 | 0.57 |
| SiT-XL$_{CFG=1.5, Sampler=ODE}$ (Ma et al., 2024) | 1,400 | 2.11 | **4.62** | 256.0 | **0.81** | **0.61** |
| SiT-XL$_{CFG=1.5, Sampler=ODE}$ + OAT-FM | +5 | **2.05** | **4.62** | **259.4** | 0.80 | **0.61** |
| SiT-XL$_{CFG=2.5, Sampler=ODE}$ | 1,400 | 6.91 | 6.42 | 391.5 | **0.89** | 0.47 |
| SiT-XL$_{CFG=2.5, Sampler=ODE}$ + OAT-FM | +5 | **6.57** | **5.98** | **394.8** | **0.89** | **0.49** |
| SiT-XL$_{CFG=1.5, Sampler=SDE}$ | 1,400 | 2.05 | 4.50 | 269.6 | **0.82** | **0.59** |
| SiT-XL$_{CFG=1.5, Sampler=SDE}$ + OAT-FM | +5 | **2.00** | **4.43** | **275.1** | **0.82** | **0.59** |
| SiT-XL$_{CFG=2.5, Sampler=SDE}$ | 1,400 | 7.75 | 6.64 | 405.0 | **0.90** | 0.45 |
| SiT-XL$_{CFG=2.5, Sampler=SDE}$ + OAT-FM | +5 | **7.44** | **5.77** | **409.9** | **0.90** | **0.46** |

**Remark.** Our OAT-based coupling improves EDM performance, whereas the classic OT-based coupling results in catastrophic performance degradation. This phenomenon reveals the essential difference between OT and OAT in the velocity smoothness. In particular, EDM learns a score function under the assumption that the noise at time $t$ is independent and isotropic Gaussian noise. When reformulating EDM as an FM model (See Appendix D.1), this assumption means that the velocity field is smoothed (Song et al., 2021b). The OT-based coupling, however, achieves static, non-Markovian transport between noise and data in the sample space, without any smoothness constraint on the velocity. In contrast, the OAT-based coupling matches the "sample-velocity" tuples at $t = 0$ with those at $t = 1$ in the product space $\mathcal{X} \times \mathcal{V}$, solving the acceleration minimization problem in (4) equivalently. The objective of (4) reveals the isotropic Gaussian prior of the acceleration, which leads to smoothed velocity (Benamou et al., 2019). Consequently, our OAT-based coupling follows the smoothness assumption and is suitable for refining EDM.

### 4.3 LARGE-SCALE CONDITIONAL IMAGE GENERATION

To verify the feasibility of OAT-FM as a phase-2 training method in practice, given the state-of-the-art image generator SiT-XL (Ma et al., 2024) trained on the ImageNet 256×256 benchmark (Deng et al., 2009), we apply OAT-FM to continually refine the model on the dataset and test it in conditional image generation tasks. We compare the model refined by OAT-FM with the original SiT-XL and other image generators on different metrics, including FID, spatial FID (sFID), Inception Score (IS) (Salimans et al., 2016), and the precision and recall measuring how well the real and generated data manifolds are overlapped with each other. In the inference phase, we apply an ODE sampler and a SDE one, respectively, to generate images with different Classifier-Free Guidance (CFG) scales.

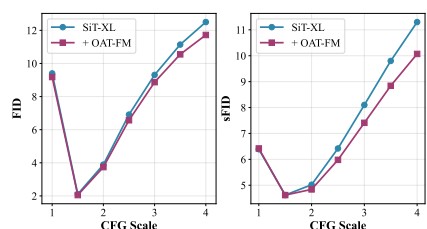

Figure 3: The comparison of *SiT-XL* and *SiT-XL + OAT-FM* on FID and sFID.

As detailed in Table 5, for the original SiT-XL model derived by 1,400 training epochs, refining it by OAT-FM with merely five more training epochs (48K batches) leads to consistent improvements in FID, sFID, and IS. The precision and recall remain stable. In Figure 3, we plot the performance of SiT-XL with and without OAT-FM while varying the CFG scale from 1.0 (no guidance) to 4.0. The results confirm that OAT-FM delivers consistent improvements across the entire range of scales.

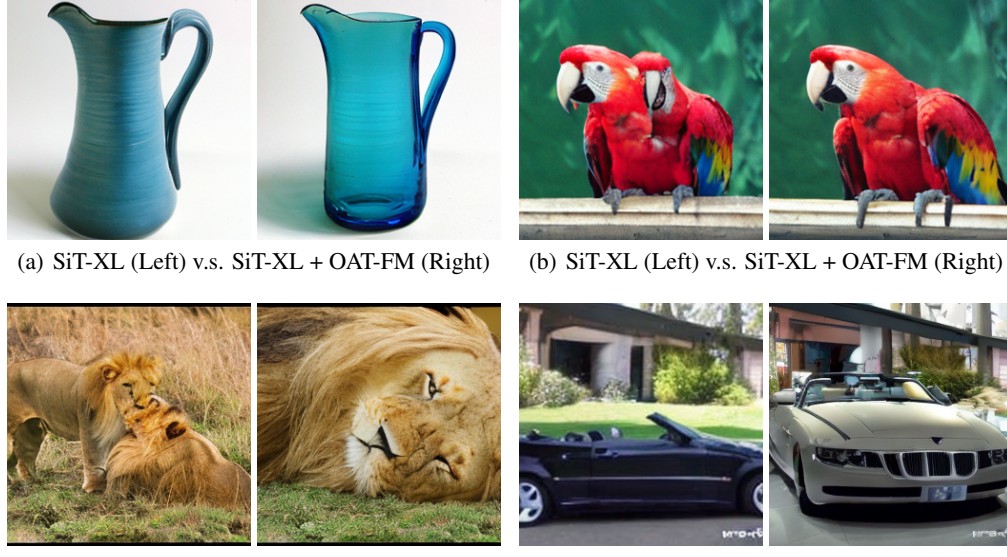

(a) SiT-XL (Left) v.s. SiT-XL + OAT-FM (Right)  (b) SiT-XL (Left) v.s. SiT-XL + OAT-FM (Right)

(c) SiT-XL (Left) v.s. SiT-XL + OAT-FM (Right)  (d) SiT-XL (Left) v.s. SiT-XL + OAT-FM (Right)

Figure 4: The visual comparison for *SiT-XL* and *SiT-XL + OAT-FM* when CFG is 4.0.

Note that, setting a large CFG scale helps generate high-quality images in general (Ma et al., 2024; Peebles & Xie, 2023) (although leading to relatively high FID/sFID scores). The superiority of OAT-FM is significant when CFG is 4.0, which demonstrates the effectiveness of OAT-FM in practice.

Interestingly, when starting from the same noise, in some cases (e.g., the "Bottle" and "Parrot" shown in Figure 4), applying OAT-FM improves image details, maintains the main semantic and spatial content created by SiT-XL, and suppresses hallucination. However, in the other cases (e.g., "Lion" and "Car" shown in Figure 4), applying OAT-FM results in the model producing images that are entirely different from those generated by SiT-XL. This phenomenon indicates that the coupling associated with OAT-FM differs from that in the original model. For flow trajectories whose endpoints obey the data distribution well, OAT-FM straightens them with almost the same endpoints. For flow trajectories whose endpoints are undesired (e.g., in the "Lion" generated by SiT-XL), OAT-FM significantly changes their directions, leading to new endpoints that may better fit the data distribution. More visual results are in Appendix D.4.2.

## 5 CONCLUSION

This work reconsiders flow matching through second-order transport, namely, optimal acceleration transport (OAT). OAT lifts the dynamics from continuity on the sample space to the Vlasov conservation law on the product space of position and velocity. The resulting product-space coupling aligns endpoints' directions and speeds, then suppresses total bending, which implies the necessary and sufficient condition for straightness. A new two-phase FM paradigm is developed, which first obtains reliable velocities using any standard flow matching/diffusion procedure and then fine-tune the model by OAT-FM. Experiments demonstrate that OAT-FM helps improve various FM models consistently, which leads to promising image generation results.

**Limitations and future work.** There are natural directions to refine the method. OAT-FM benefits from reasonably accurate endpoint velocities, so currently, training from scratch via OAT-FM can be fragile due to its dependence on velocity information — early velocity estimates are noisy and can misguide the product-space coupling. Warm starts using CFM or self-distillation in the spirit of Shortcut (Frans et al., 2025) and consistency models (Kim et al., 2024) may offer simple remedies before handing off to OAT-FM. From a computational standpoint, mini-batch couplings scale quadratically in batch size. In the future, we plan to explore the dual form of OAT problem and develop a more efficient OAT solver to accelerate OAT-FM further.

## Ethics Statement

The datasets used in this paper are all publicly available and do not involve any ethical issues.

## Reproducibility Statement

We are committed to ensuring the reproducibility of our research. The source code is provided in the supplementary materials, which can be used to reproduce the main results presented in this paper.

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

APPENDIX

## A  USE OF LARGE LANGUAGE MODELS (LLMS)

In this work, LLMs are used for language refinement of the paper. We also use LLMs to assist a part of code implementation.

## B  DETAILED PROOFS

Throughout this paper, we take absolutely continuous paths $x : [0, 1] \to \mathbb{R}^d$ with $v = \dot{x}$ and $a = \dot{v}$. For $\|v(t)\| > 0$, define the unit direction

$$\mathbf{s}(t) := \frac{v(t)}{\|v(t)\|}, \tag{15}$$

and decompose the acceleration into tangential (speed) and normal (bending) parts

$$a_{\parallel}(t) := (\mathbf{s}(t) \cdot a(t)) \, \mathbf{s}(t), \qquad a_{\perp}(t) := a(t) - a_{\parallel}(t). \tag{16}$$

These obey

$$\frac{d}{dt} \|v(t)\| = \mathbf{s}(t) \cdot a(t), \qquad \dot{\mathbf{s}}(t) = \frac{a_{\perp}(t)}{\|v(t)\|}. \tag{17}$$

Thus $a_{\parallel}$ modulates speed, while $a_{\perp}$ is the source of bending. In particular, $a_{\perp}(t) \equiv 0$ iff $\mathbf{s}(t)$ is constant, hence $x(t)$ is straight. When needed, the instantaneous curvature magnitude is

$$\|\dot{\mathbf{s}}(t)\| = \frac{\|a_{\perp}(t)\|}{\|v(t)\|}. \tag{18}$$

### B.1  PROOF OF PROPOSITION 1

Assume first that the trajectory is straight. Then there exist a unit vector $u$ and a scalar function $s(t)$ such that[3] such that

$$x(t) = x_0 + s(t) \, u, \qquad t \in [0, 1]. \tag{19}$$

Differentiating gives

$$v(t) = \dot{s}(t) \, u, \qquad a(t) = \ddot{s}(t) \, u. \tag{20}$$

Wherever $\|v(t)\| > 0$, the unit direction of motion equals the fixed $u$, so $\mathbf{s}(t) = u$ is constant in time, and $a(t)$ is a scalar multiple of $v(t)$. At instants where $v(t) = 0$, the direction is immaterial and the collinearity $a(t) \parallel v(t)$ is trivially satisfied. In the decomposition above this means $a_{\perp}(t) = 0$ whenever $\|v(t)\| > 0$.

Conversely, suppose the velocity direction is time invariant and the acceleration is parallel to the velocity. Then there exist a unit vector $u$ and scalar functions $\alpha(t)$ and $\beta(t)$ such that

$$v(t) = \alpha(t) \, u, \qquad a(t) = \beta(t) \, u, \qquad t \in [0, 1]. \tag{21}$$

From $\dot{v} = a$ it follows that $\dot{\alpha}(t) = \beta(t)$. Integrating $v = \dot{x}$ yields $x(t) = x_0 + (\int_0^t \alpha(\tau) d\tau) u$, which lies on the fixed line $x_0 + \mathbb{R}u$. In the notation introduced before the proposition, $\dot{\mathbf{s}}(t) = a_{\perp}(t)/\|v(t)\|$, hence $a_{\perp}(t) \equiv 0$ implies $\mathbf{s}(t)$ is constant wherever $\|v(t)\| > 0$, which is the same conclusion.

Finally, when $a \equiv 0$ we have $\dot{v} = 0$ so $v(t) \equiv v_0$, and therefore

$$x(t) = x_0 + t \, v_0, \tag{22}$$

which is straight motion at constant speed, as in the first–order Benamou–Brenier setting.

---

[3]Here $u$ denotes the fixed direction of the line, and $s(t)$ is the scalar coordinate along that line: writing $x(t) = x_0 + s(t)u$ exactly encodes that $x(t) \in x_0 + \mathbb{R}u$. This is different from $\mathbf{s}(t) = v(t)/\|v(t)\|$, which is the instantaneous direction of motion. In the straight case one has $v(t) = \dot{s}(t) \, u$, hence $\mathbf{s}(t) = v(t)/\|v(t)\| = \operatorname{sign}(\dot{s}(t)) \, u$ wherever $\|v(t)\| > 0$. Thus $\mathbf{s}(t) = u$ when $\dot{s}(t) > 0$ and $\mathbf{s}(t) = -u$ when $\dot{s}(t) < 0$; any sign change can occur only at instants where $v(t) = 0$ (where $\mathbf{s}$ is undefined), which is why direction constancy is stated on the set $\{t : \|v(t)\| > 0\}$.

## B.2 Proof of Theorem 2

Before proving Theorem 2, we first consider straightening a single trajectory by solving an acceleration minimization problem, which leads to the following theorem.

**Theorem 4 (Straightening a single trajectory via acceleration minimization).** *Let $(x_0, v_0)$ and $(x_1, v_1)$ be two points in a product space $\mathcal{X} \times \mathcal{V}$, where $\mathcal{X}$ denotes a sample space and $\mathcal{V}$ denotes a velocity space. Among all twice differentiable trajectories $x(t) : [0,1] \to \mathbb{R}^d$ taking them as their endpoints, the acceleration minimization problem, i.e.,*

$$\min_a \frac{1}{2} \int_0^1 \|a(t)\|^2 \, dt, \qquad s.t. \ v_0 + \int_0^1 a(t) dt = v_1, \ and \ x_0 + \int_0^1 \int_0^t a(s) ds dt = x_1, \quad (23)$$

*that admits a unique coordinate-wise cubic interpolation minimizer. Moreover,*

1. *Solving the problem leads to a straight trajectory with constant velocity (i.e., $a \equiv 0$) is feasible if and only if $v_0 = v_1$ and they are collinear with $x_1 - x_0$.*

2. *Solving the problem leads to a straight trajectory (i.e., $s(t)$ is constant and $a_\perp(t) \equiv 0$) if and only if $v_0$ and $v_1$ are collinear with $x_1 - x_0$.*

3. *Otherwise, it bends exactly to match the endpoints' orthogonal components.*

*Proof.* Existence and uniqueness follow from strict convexity on $\mathcal{H}^2([0,1]; \mathbb{R}^d)$ with the stated boundary constraints.[4] The Euler–Lagrange equation is $x^{(4)}(t) = 0$ in each coordinate with four boundary conditions, hence the unique solution is a cubic polynomial in each coordinate, i.e., the cubic interpolation determined by $(x_0, v_0)$ and $(x_1, v_1)$.

For straightness when feasible, let $u := x_1 - x_0 \neq 0$ and choose a unit vector $e$ with $u = \|u\|e$. Suppose $v_0 = \alpha_0 e$ and $v_1 = \alpha_1 e$. Decompose any admissible curve as

$$x(t) = x^\|(t)\, e + x^\perp(t) \quad \text{with} \quad x^\perp(t) \perp e. \quad (24)$$

The boundary data enforce

$$x^\perp(0) = x^\perp(1) = 0 \quad \text{and} \quad \dot{x}^\perp(0) = \dot{x}^\perp(1) = 0. \quad (25)$$

The cost splits orthogonally,

$$\int_0^1 \|\ddot{x}\|^2 \, dt = \int_0^1 \|\ddot{x}^\|(t)\|^2 \, dt + \int_0^1 \|\ddot{x}^\perp(t)\|^2 \, dt, \quad (26)$$

so $x^\perp$ solves a homogeneous strictly convex problem with these boundary data. The Euler–Lagrange equation $(x^\perp)^{(4)} = 0$ forces all cubic coefficients to vanish, hence $x^\perp \equiv 0$. The minimizer is therefore straight, of the form $x(t) = x_0 + s(t)e$ with $v(t) = \dot{s}(t)e$. Wherever $\|\dot{x}(t)\| > 0$ the direction $\mathbf{s}(t) = \dot{x}(t)/\|\dot{x}(t)\|$ equals $e$ and is constant, and with

$$a_\|(t) = (\mathbf{s}(t) \cdot a(t))\, \mathbf{s}(t), \qquad a_\perp(t) = a(t) - a_\|(t), \quad (27)$$

one has $a_\perp(t) \equiv 0$.

For general endpoints, choose an orthonormal basis whose first axis is $e = u/\|u\|$ if $u \neq 0$ (otherwise any orthonormal basis). In this, the functional and constraints separate across coordinates; each coordinate solves the scalar cubic interpolation with its own boundary data. The perpendicular coordinates are uniquely determined by the perpendicular parts of $(x_0, v_0)$ and $(x_1, v_1)$ and vanish exactly in the straightness–feasible case above. Thus, the minimizer bends only to the extent required by the endpoint data.

Finally, relate the acceleration objective to a straightening proxy. With $u = x_1 - x_0$ and using $v = \dot{x}, a = \dot{v}$,

$$v(t) - u = \int_0^t a(s) \, ds - \int_0^1 (1-s)\, a(s) \, ds. \quad (28)$$

---

[4] $\mathcal{H}$ denote the Hilbert space.

Hence $v - u$ is a bounded linear image of $a$ on $L^2([0, 1]; \mathbb{R}^d)$. Poincaré–type estimates give constants $c_1, c_2 > 0$, independent of the endpoints, such that for every admissible curve

$$c_1 \int_0^1 \|a(t)\|^2 \, dt \le \int_0^1 \|v(t) - u\|^2 \, dt + \|v_1 - v_0\|^2 \le c_2 \int_0^1 \|a(t)\|^2 \, dt. \tag{29}$$

Thus, minimizing (23) controls and optimizes the straightening regression loss. $\qquad\square$

Essentially, Theorem 2 extends Theorem 4 to a distributional scenario. By the static–dynamic equivalence for the acceleration cost at horizon $T = 1$, the dynamic OAT value equals

$$\min_{\pi \in \Pi(\mu_0, \mu_1)} \frac{1}{2} \mathbb{E}_{(z_0, z_1) \sim \pi} \Big[ c_A^2(z_0, z_1) \Big],$$

hence there exists an optimal coupling $\pi^* \in \Pi(\mu_0, \mu_1)$. For each endpoint pair $(z_0, z_1) = (x_0, v_0; x_1, v_1)$ in the support of $\pi^*$, the single–path problem with boundary data $(x_0, v_0) \to (x_1, v_1)$ has a unique minimizer, namely the coordinatewise cubic (Y. Chen interpolation). Let

$$\Phi_t(z_0, z_1) := \big( x_{z_0, z_1}(t), \, v_{z_0, z_1}(t) \big) \quad \text{for} \quad t \in [0, 1].$$

Define $\mu_t := \Phi_{t \#} \pi^*$, we then have $\mu_0 = \Phi_{0 \#} \pi^* = \mu_0$ and $\mu_1 = \Phi_{1 \#} \pi^* = \mu_1$ by the marginal constraints on $\pi^*$, and by construction the family $(\mu_t)_{t \in [0,1]}$ is obtained by transporting the coupling along characteristics that solve $\dot{x} = v$ and $\dot{v} = a$. This representation satisfies the kinetic continuity equation in the distributional sense and attains the minimum action.

Finally, fix $(z_0, z_1)$ in the support of $\pi^*$ and consider its cubic characteristic $t \mapsto (x_{z_0, z_1}(t), v_{z_0, z_1}(t))$. By Theorem 4, this trajectory is straight if and only if $v_0$ and $v_1$ are collinear with $u := x_1 - x_0$; in that case $\mathbf{s}(t) = v(t)/\|v(t)\|$ is constant wherever $\|v(t)\| > 0$ and $a_\perp(t) \equiv 0$. Otherwise, the trajectory bends exactly to match the endpoints' orthogonal components. Since this holds for every $(z_0, z_1)$ in the support of $\pi^*$, the corollary follows.

### B.3 PROOF OF THEOREM 3

Fix $t \in [0, 1]$ with endpoint pair $z_0 = (x_0, v_0)$ and $z_1 = (x_1, v_1)$, and redefine the variables

$$u := x_1 - x_0, \qquad \bar{v} := \frac{v_0 + v_1}{2}, \qquad w := v_1 - v_0, \qquad v_t := v_\theta(x_t, t). \tag{30}$$

The per–sample integrand of (7) is

$$\ell_A(v_t, \alpha) := \alpha \left( \left\| \frac{v_0 + v_t}{2} - u \right\|^2 + \left\| \frac{v_t + v_1}{2} - u \right\|^2 \right) + (1 - \alpha) \left( \|v_t - v_0\|^2 + \|v_1 - v_t\|^2 \right). \tag{31}$$

We first rewrite the two pairs of squares by completing the square. Using the identity

$$\|x + a\|^2 + \|x + b\|^2 = \frac{1}{2} \|2x + (a + b)\|^2 + \frac{1}{2} \|a - b\|^2,$$

with $x = \frac{1}{2} v_t$ and $(a, b) = (\frac{1}{2} v_0 - u, \frac{1}{2} v_1 - u)$, we obtain

$$\left\| \frac{v_0 + v_t}{2} - u \right\|^2 + \left\| \frac{v_t + v_1}{2} - u \right\|^2 = \frac{1}{2} \|v_t - (2u - \bar{v})\|^2 + \frac{1}{8} \|w\|^2, \tag{32}$$

and similarly, we have

$$\|v_t - v_0\|^2 + \|v_1 - v_t\|^2 = 2 \|v_t - \bar{v}\|^2 + \frac{1}{2} \|w\|^2. \tag{33}$$

Substituting these into $\mathcal{L}_\alpha$ yields to

$$\mathcal{L}_\alpha(v_t) = \frac{\alpha}{2} \|v_t - (2u - \bar{v})\|^2 + 2(1 - \alpha) \|v_t - \bar{v}\|^2 + \left( \frac{1}{2} - \frac{3}{8} \alpha \right) \|w\|^2. \tag{34}$$

Next, minimize over $v_t$. For $p, q > 0$ and $a, b \in \mathbb{R}^d$,

$$\min_x \left\{ p \|x - a\|^2 + q \|x - b\|^2 \right\} = \frac{pq}{p + q} \|a - b\|^2.$$

Applying this with $p = \frac{\alpha}{2}$, $q = 2(1-\alpha)$, $a = 2u - \bar{v}$, $b = \bar{v}$, and $\|a - b\|^2 = 4\|u - \bar{v}\|^2$, yields

$$\min_{v_t} \mathcal{L}_\alpha(v_t) = \frac{8\alpha(1-\alpha)}{4-3\alpha} \|u - \bar{v}\|^2 + \left(\frac{1}{2} - \frac{3}{8}\alpha\right)\|w\|^2. \tag{35}$$

For all $\alpha \in [0,1]$ one has $\frac{1}{12}\frac{8\alpha(1-\alpha)}{4-3\alpha} < \frac{1}{2} - \frac{3}{8}\alpha$, and hence

$$\frac{8\alpha(1-\alpha)}{4-3\alpha}\|u - \bar{v}\|^2 + \left(\frac{1}{2} - \frac{3}{8}\alpha\right)\|w\|^2 \geq \frac{1}{12}\frac{8\alpha(1-\alpha)}{4-3\alpha}\left(12\|u - \bar{v}\|^2 + \|w\|^2\right). \tag{36}$$

Equality holds if and only if $\|w\| = 0$ (i.e., $v_1 = v_0$). Exact equality at the pair level occurs precisely when $v_1 = v_0$, and minimizing over $v_t$ yields to

$$v_t^\star = \arg\min_x \left\{ \frac{\alpha}{2}\|x - (2u - \bar{v})\|^2 + 2(1-\alpha)\|x - \bar{v}\|^2 \right\} = \frac{2\alpha}{4-3\alpha}\,u + \frac{4-5\alpha}{4-3\alpha}\bar{v}. \tag{37}$$

The single–pair OAT cost for unit horizon is $c_\mathcal{A}^2(z_0, z_1) = 12\|u - \bar{v}\|^2 + \|w\|^2$. Therefore, we have

$$\min_{v_t} \mathcal{L}_\alpha(v_t) \geq c(\alpha)\, c_\mathcal{A}^2(z_0, z_1), \tag{38}$$

with $c(\alpha) := \min\{\frac{\alpha(1-\alpha)}{6-\frac{9}{2}\alpha}, \frac{1}{2} - \frac{3}{8}\alpha\}$, with $c(\alpha)$ attains its maximum at $\alpha = \frac{2}{3}$ when $\alpha \in [0,1]$, with $c(\frac{2}{3}) = \frac{2}{27}$. Hence, with $\alpha = \frac{2}{3}$, we have $\mathcal{L}_{2/3}(v_t) \geq \frac{2}{27} c_\mathcal{A}^2(z_0, z_1), \forall v_t$. Finally, average over $t \sim \text{Unif}[0,1]$ and $(z_0, z_1) \sim \pi$, then minimize over $\pi \in \Pi(\mu_0, \mu_1)$ and over $\theta$, to obtain $\mathcal{L}_{\text{OAT}}(\mu_0, \mu_1) \geq \frac{2}{27} \mathcal{A}_2^2(\mu_0, \mu_1)$.

# C    LEARNING ALGORITHM

## C.1    DERIVATION OF (9)

In the context of FM, we have $\pi(z_0, z_1) = \pi(x_0, v_0, x_1, v_1) = \pi_x(x_0, x_1)\pi(v_0|x_0)\pi(v_1|x_1)$, where $\pi_x(x_0, x_1) \in \Pi(\rho_0, \rho_1)$ is the marginal coupling corresponding to sample pairs and $\pi(\cdot|x_t) = \delta_{v_\theta(x_t, t)}$ is the Dirac measure determined by the flow model $v_\theta$. Based on the decomposition that $\pi(z_0, z_1) = \pi_x(x_0, x_1)\delta_{v_\theta(x_0, 0)}(v_0)\delta_{v_\theta(x_1, 1)}(v_1)$, we can merely optimize $\pi_x(x_0, x_1)$ and reformulate the lower-level problem in (7) as

$$\min_{\pi \in \Pi(\mu_0, \mu_1)} \mathbb{E}_{(z_0, z_1) \sim \pi}\left[c_\mathcal{A}^2(z_0, z_1)\right]$$

$$= \min_{\pi \in \Pi(\mu_0, \mu_1)} \mathbb{E}_{(z_0, z_1) \sim \pi}\left[12\left\|\frac{x_1 - x_0}{T} - \frac{v_1 + v_0}{2}\right\|_2^2 + \|v_1 - v_0\|_2^2\right]$$

$$\leq \min_{\pi_x \in \Pi(\rho_0, \rho_1)} \mathbb{E}_{(x_0, x_1) \sim \pi_x}\left[12\left\|\frac{x_1 - x_0}{T} - \underbrace{\frac{v_\theta(x_1, 1) + v_\theta(x_0, 0)}{2}}_{\text{Denoted as } \bar{v}_{x_0, x_1}}\right\|_2^2 + \|\underbrace{v_\theta(x_1, 1) - v_\theta(x_0, 0)}_{\text{Denoted as } \tilde{v}_{x_0, x_1}}\|_2^2\right]$$

$$\left(\text{When considering the decomposition } \pi(z_0, z_1) = \pi_x(x_0, x_1)\delta_{v_\theta(x_0, 0)}(v_0)\delta_{v_\theta(x_1, 1)}(v_1)\right)$$

$$= \min_{\pi_x \in \Pi(\rho_0, \rho_1)} \mathbb{E}_{(x_0, x_1) \sim \pi_x}\left[12\|x_1 - x_0 - \bar{v}_{x_0, x_1}\|_2^2 + \|\tilde{v}_{x_0, x_1}\|_2^2\right] \quad (\text{set } T = 1).$$

The inequality in the above derivation is because we impose the decomposable structure on $\pi$, which shrinks its feasible domain from $\Pi(\mu_0, \mu_1)$ to $\Pi'(\mu_0, \mu_1) := \{\pi \mid \pi(z_0, z_1) = \pi_x(x_0, x_1)\delta_{v_\theta(x_0, 0)}(v_0)\delta_{v_\theta(x_1, 1)}(v_1), \pi_x = \iint_{v_0, v_1} \pi, \text{ and } \pi \in \Pi\}$.

Suppose that we have a batch of samples with size $B$, i.e., $\{x_{1,i}\}_{i=1}^B \sim \mathcal{D}$, and a batch of noise with the same size, i.e., $\{x_{0,i}\}_{i=1}^B \sim \mathcal{N}(0, I)$. We can get their velocities, i.e., $\{v_{0,i} \leftarrow v_\theta(x_{0,i}, 0)\}_{i=1}^B$ and $\{v_{1,i} \leftarrow v_\theta(x_1, 1)\}_{i=1}^B$. The above problem can be rewritten in a discrete format:

$$\arg\min_{\mathbf{T}}\langle \mathbf{C}, \mathbf{T}\rangle, \quad \text{s.t. } \mathbf{T}\mathbf{1}_B = \frac{1}{B}\mathbf{1}_B, \mathbf{T}^\top\mathbf{1}_B = \frac{1}{B}\mathbf{1}_B, \tag{39}$$

Table 6: Configurations of the flow/diffusion models used in our experiments.

| Model | Paradigm | Objective | Path / Coupling | Architecture | Space |
|---|---|---|---|---|---|
| **EDM** (Karras et al., 2022) | Diffusion | Score Matching | VP/VE Noise Schedule | U-Net | Pixel |
| **FM** (Lipman et al., 2023) | Flow | Flow Matching | Gaussian-Data Path | U-Net | Pixel |
| **I-CFM** (Tong et al., 2024) | Flow | Conditional FM | Linear Path (Independent) | U-Net | Pixel |
| **VP-CFM** (Albergo et al., 2023) | Flow | Conditional FM | Trigonometric Path | U-Net | Pixel |
| **SB-CFM** (Tong et al., 2024) | Flow | Conditional FM | Brownian Bridge Path | U-Net | Pixel |
| **OT-CFM** (Tong et al., 2024) | Flow | Conditional FM | Linear Path (OT Coupling) | U-Net | Pixel |
| **SiT-XL** (Ma et al., 2024) | Flow | Velocity Matching | Linear / GVP Path | DiT | Latent |

where $\langle \cdot, \cdot \rangle$ denotes inner product, $\mathbf{T}$ is the coupling matrix, and $\mathbf{C} = [c_{ij}] \in \mathbb{R}^{B \times B}$ is the cost matrix, whose element $c_{ij} = 12 \| x_{1,i} - x_{0,j} - \frac{v_{0,j} + v_{1,i}}{2} \|_2^2 + \| v_{1,i} - v_{0,j} \|_2^2$. Following the OT-CFM rationale, the problem can be cast as a linear program with computational complexity $\mathcal{O}(B^3 \log \|\mathbf{C}\|_\infty)$. The optimizer can be further approximated by adding an entropic regularizer of $\mathbf{T}$ weighted by $\epsilon$, i.e., $\epsilon \langle \mathbf{T}, \log \mathbf{T} \rangle$. This allows the problem to be solved efficiently by Sinkhorn method, whose complexity is $\mathcal{O}(B^2 \log B)$, and the exact OT result can be recovered by taking $\epsilon$ sufficiently small. We refer to Peyré & Cuturi (2019) for further details.

### C.2 THE SCHEME OF LEARNING ALGORITHM

Algorithm 1 provides the algorithmic scheme of OAT-FM. This algorithm works as the phase-2 training step in our two-phase FM paradigm.

---

**Algorithm 1** OAT-FM

---

**Require:** A phase-1 pre-trained model $v_{\theta_0}$, dataset $\mathcal{D}$, EMA decay rate $\lambda$, batch size $B$
**Ensure:** Refined velocity field $v_\theta$
1: **Initialize** $v_\theta \leftarrow v_{\theta_0}$
2: **while** training **do**
3:      Sample a batch with size $B$: $\{x_{1,i}\}_{i=1}^B \sim \mathcal{D}$, $\{x_{0,i}\}_{i=1}^B \sim \mathcal{N}(0, I)$, and $t \sim \mathcal{U}[0, 1]$
4:      $\{v_{0,i} \leftarrow v_\theta(x_{0,i}, 0)\}_{i=1}^B$, $\{v_{1,i} \leftarrow v_\theta(x_1, 1)\}_{i=1}^B$, and $\{\bar{v}_{ij} \leftarrow v_{0,j} + v_{1,j}\}_{i,j=1}^B$
5:      Compute the optimal coupling matrix $\mathbf{T}^*$ by solving (39).
6:      Sampling $K$ pairs $(x_1, x_0) \sim \mathbf{T}^*$ and obtain the corresponding $v_0$ and $v_1$.
7:      $x_t \leftarrow (1 - t)x_0 + tx_1$, $v_t \leftarrow v_\theta(x_t, t)$, and compute $\mathcal{L}_{\text{OAT}}$ accordingly.
8:      Update model: $\theta' \leftarrow \theta - \nabla_\theta \mathcal{L}_{\text{OAT}}$
9:      $\theta \leftarrow \text{stopgrad}(\lambda\theta + (1 - \lambda)\theta')$
10: **end while**

---

## D    DETAILED EXPERIMENT SETTINGS

### D.1    CONVERSION FROM VP/VE MODELS TO FLOW MATCHING

Following the work in (Lee et al., 2024), our training process begins with a pre-trained diffusion model. Specifically, we use models trained with the Elucidating the Design Space of Diffusion-Based Generative Models (EDM) framework (Karras et al., 2022), which provides a unified perspective on score-based models including Variance Preserving (VP) and Variance Exploding (VE) SDEs. To adapt this model for flow matching, we re-parameterize its input and output to function as a velocity field predictor. This allows us to initialize our model with a strong, pre-trained diffusion backbone, facilitating a seamless transition to the second-phase OAT-FM training paradigm.

**EDM Preconditioning**. The EDM model takes a noise-corrupted input $x_\sigma = x_1 + \sigma z$ (where $z \sim \mathcal{N}(0, I)$) and a noise level $\sigma$, is trained to denoise it. Here, the noise at time $t$ is assumed to be isotropic Gaussian and independent of the data. Accordingly, the forward process of EDM is Markovian, implying a smooth score function and a smooth velocity field (Song et al., 2021b). The denoised output $D_\theta(x_\sigma, \sigma)$ is an estimate of the original data $x_1$ and is formulated using a

preconditioning scheme:

$$D_\theta(x_\sigma, \sigma) = c_{\text{skip}}(\sigma)x_\sigma + c_{\text{out}}(\sigma)F_\theta(c_{\text{in}}(\sigma)x_\sigma, c_{\text{noise}}(\sigma)), \tag{40}$$

where the scaling factors $c_{\text{skip}}, c_{\text{out}}, c_{\text{in}}$, and the time embedding $c_{\text{noise}}$ are functions of $\sigma$ designed to improve network conditioning and training stability. Specifically, they are defined as

$$c_{\text{skip}}(\sigma) = \frac{\sigma_{\text{data}}^2}{\sigma^2 + \sigma_{\text{data}}^2}, \quad c_{\text{out}}(\sigma) = \frac{\sigma\sigma_{\text{data}}}{\sqrt{\sigma^2 + \sigma_{\text{data}}^2}}, \quad c_{\text{in}}(\sigma) = \frac{1}{\sqrt{\sigma_{\text{data}}^2 + \sigma^2}}, \quad c_{\text{noise}}(\sigma) = \frac{1}{4}\log(\sigma),$$

where $\sigma_{\text{data}}$ is the standard deviation of the training data.

**Flow Matching Objective**. In the flow matching formulation, we aim to learn a velocity field $v_\theta(x, t)$ that models the linear trajectory between a data sample $x_1$ and a noise sample $x_0 \sim \mathcal{N}(0, I)$. The path is defined as $x_t = (1 - t)x_0 + tx_1$ for $t \in [0, 1]$. The ground truth velocity is simply $v(x_t, t) = x_1 - x_0$. The model is trained by minimizing the following L2 loss:

$$\mathcal{L}_{\text{FM}} = \mathbb{E}_{x_0, x_1, t}\left[\|v_\theta(x_t, t) - (x_1 - x_0)\|_2^2\right]. \tag{41}$$

**Conversion to Velocity Field**. To convert the pre-trained denoiser $D_\theta$ into a velocity predictor $v_\theta$, we use an adapter that maps the flow matching inputs $(x_t, t)$ to the diffusion model's expected inputs $(x_\sigma, \sigma)$. This mapping is defined by:

$$\sigma(t) = \frac{1 - t}{t}, \quad x_\sigma = \frac{x_t}{t} = \frac{(1 - t)x_0 + tx_1}{t} = x_1 + \frac{1 - t}{t}x_1 = x_1 + \sigma(t)x_0. \tag{42}$$

This transformation effectively converts the point $x_t$ on the linear interpolation path into a correctly scaled noisy sample $x_\sigma$ that the EDM model can process. We then feed $x_\sigma$ and $\sigma(t)$ into the pre-trained EDM denoiser to obtain an estimate of the clean data, $\hat{x}_1 = D_\theta(x_\sigma, \sigma(t))$.

Finally, we construct the velocity prediction $v_\theta(x_t, t)$ from this estimate $\hat{x}_1$. Based on the definition of $x_t$, we have $x_1 - x_0 = (x_1 - x_t)/(1 - t)$. By substituting our estimate $\hat{x}_1$ for the true $x_1$, we obtain our velocity field parameterization:

$$v_\theta(x_t, t) = \frac{\hat{x}_1 - x_t}{1 - t} = \frac{D_\theta\left(\frac{x_t}{t}, \frac{1-t}{t}\right) - x_t}{1 - t}. \tag{43}$$

### D.2 LOW-DIMENSIONAL OT BENCHMARK

**Dataset**. Following the prior work in (Tong et al., 2024), we evaluate performance on five two-dimensional distribution mapping tasks. These benchmarks test the ability of a model to learn a transport map from a source distribution to the target distribution. The specific pairs are: $i$) a standard Gaussian to a mixture of 8 Gaussians ($\mathcal{N} \to$ 8gs), $ii$) a standard Gaussian to two interleaved moons ($\mathcal{N} \to$ moons), $iii$) a standard Gaussian to an S-shaped curve ($\mathcal{N} \to$ scurve), $iv$) moons to 8 Gaussians (moons $\to$ 8gs), and $v$) 8 Gaussians to moons (8gs $\to$ moons).

**Training**. For all experiments, the vector field $v_\theta$ is parameterized by a standard Multi-Layer Perceptron (MLP) that accepts concatenated position and time vectors as input. The MLP consists of 3 hidden layers, each with a width of 64 neurons, followed by a SELU activation function. The final layer maps the representation back to a 2-dimensional vector, representing the velocity at the given point in space-time. First, we train the baseline models, i.e., FM (Lipman et al., 2023), I/SB/OT-CFM (Tong et al., 2024), and VP-CFM (Albergo et al., 2023), for 20,000 batches (each batch contains 256 data points). Subsequently, each of these pre-trained models is refined using our OAT-FM for an additional 20,000 batches. During the OAT-FM training, we employ a slowly-updating strategy. The OAT-FM is updated with a hard copy of the online model's weights every 500 batches. For the OAT-FM objective, we set the balancing hyperparameter $\alpha$ to 0.70.

**Evaluation**. We assess model performance using two key metrics. First, to measure the quality of the learned terminal distribution, we compute the 2-Wasserstein distance ($\mathcal{W}_2$) between the generated samples and the true target samples. Second, to evaluate the efficiency of the learned transport path, we use the Normalized Path Energy (NPE) defined in (14). An NPE value near zero indicates that the learned path is close to the dynamic optimal transport plan. For both metrics, we use a test set of 1,024 samples. Trajectories are generated by integrating the learned vector field from $t = 0$ to

Table 7: A comparative analysis of **one-step** generation performance, evaluating data fitting (2-Wasserstein) and optimal transport approximation (normalized path energy). We run each task in five trials and record the average performance and standard deviation.

| Task | $\mathcal{N} \to$8gs | 8gs$\to$moons | $\mathcal{N} \to$moons | $\mathcal{N} \to$scurve | moons$\to$8gs |
|---|---|---|---|---|---|
| Method | $\mathcal{W}_2^2 \downarrow$ | $\mathcal{W}_2^2 \downarrow$ | $\mathcal{W}_2^2 \downarrow$ | $\mathcal{W}_2^2 \downarrow$ | $\mathcal{W}_2^2 \downarrow$ |
| OFM | $0.71_{\pm0.27}$ | $0.22_{\pm0.02}$ | $0.22_{\pm0.01}$ | $1.99_{\pm0.25}$ | $0.46_{\pm0.11}$ |
| FM | $21.50_{\pm0.07}$ | $4.05_{\pm0.01}$ | $8.08_{\pm0.05}$ | $79.01_{\pm1.18}$ | $13.48_{\pm0.15}$ |
| +OAT-FM | $\mathbf{0.43}_{\pm0.08}$ | $\mathbf{0.14}_{\pm0.02}$ | $\mathbf{0.12}_{\pm0.02}$ | $\mathbf{1.41}_{\pm0.23}$ | $\mathbf{0.32}_{\pm0.07}$ |
| I-CFM | $23.46_{\pm0.15}$ | $4.47_{\pm0.08}$ | $7.86_{\pm0.03}$ | $79.31_{\pm1.41}$ | $14.55_{\pm0.28}$ |
| +OAT-FM | $\mathbf{0.42}_{\pm0.10}$ | $\mathbf{0.17}_{\pm0.02}$ | $\mathbf{0.12}_{\pm0.01}$ | $\mathbf{1.41}_{\pm0.26}$ | $\mathbf{0.51}_{\pm0.13}$ |
| VP-CFM | $12.17_{\pm0.29}$ | $8.46_{\pm0.25}$ | $3.70_{\pm0.05}$ | $47.73_{\pm1.94}$ | $7.51_{\pm0.21}$ |
| +OAT-FM | $\mathbf{0.45}_{\pm0.11}$ | $\mathbf{0.20}_{\pm0.03}$ | $\mathbf{0.12}_{\pm0.01}$ | $\mathbf{1.35}_{\pm0.23}$ | $\mathbf{0.36}_{\pm0.07}$ |
| SB-CFM | $0.63_{\pm0.09}$ | $0.19_{\pm0.04}$ | $0.20_{\pm0.04}$ | $1.91_{\pm0.35}$ | $0.35_{\pm0.05}$ |
| +OAT-FM | $\mathbf{0.42}_{\pm0.10}$ | $\mathbf{0.13}_{\pm0.01}$ | $\mathbf{0.10}_{\pm0.01}$ | $\mathbf{1.22}_{\pm0.26}$ | $\mathbf{0.33}_{\pm0.07}$ |
| OT-CFM | $0.48_{\pm0.07}$ | $0.19_{\pm0.05}$ | $\mathbf{0.12}_{\pm0.01}$ | $1.64_{\pm0.22}$ | $0.35_{\pm0.08}$ |
| +OAT-FM | $\mathbf{0.40}_{\pm0.08}$ | $\mathbf{0.13}_{\pm0.00}$ | $\mathbf{0.12}_{\pm0.03}$ | $\mathbf{1.23}_{\pm0.19}$ | $\mathbf{0.30}_{\pm0.06}$ |

$t = 1$ using a 4th-order Runge-Kutta (RK4) solver (Dormand & Prince, 1980) with 101 discretization steps and absolute and relative error tolerances of $10^{-6}$. The path energy integral is numerically approximated using the trapezoidal rule over the computed trajectory points.

**Comparisons in one-step generation.** Table 7 presents our experiments in the one-step generation setting. We include OFM (Kornilov et al., 2024) as an additional baseline. Here, we only compare different methods on data fitting (2-Wasserstein) because all the methods apply one-step generation. It is worth noting that OFM requires an ICNN architecture, whereas our method and the other baselines utilize the same MLP backbone. The results demonstrate the effectiveness of our method.

## D.3 CIFAR-10 32×32

### D.3.1 BASIC SETTINGS

**Dataset**. We evaluate our model on the CIFAR-10 dataset (Krizhevsky, 2009), a widely-used benchmark for image generation. The dataset consists of 60,000 color images with size 32×32 in 10 classes, partitioned into 50,000 training images and 10,000 test images.

**Training**. We refine publicly available FM, I-CFM, and OT-CFM models from `https://github.com/atong01/conditional-flow-matching/tree/main/examples/images/cifar10` that were pre-trained for 400K iterations. The neural network is a U-Net architecture (Ho et al., 2020) with an implementation adapted from the guided-diffusion repository in `https://github.com/openai/guided-diffusion`. The U-Net has a base of 128 channels, channel multipliers of $[1, 2, 2, 2]$, two residual blocks per resolution, and applies four-head self-attention at the 16×16 resolution. The boundary velocity fields required for the OAT-FM loss are estimated using a target network, which is an exponential moving average (EMA) of the online model's weights with a decay of 0.9999 (See Algorithm 1). For the OAT-FM objective, we set the balancing hyperparameter $\alpha$ to 0.75. We use the Adam optimizer with a learning rate of $2 \times 10^{-4}$, and apply gradient clipping with a maximum norm of 1.0. We use OAT-FM to train these three models with an additional 1K iterations. For EDM, we downloaded the checkpoints from `https://drive.google.com/drive/folders/18dWE-LiodXdCG0RDNegySzRnyRdcwamW`. This model utilizes the DDPM++ architecture (SongUNet) from the work in (Song et al., 2021a), which is a U-Net composed of residual blocks, self-attention, and positional timestep embeddings. Following the methodology in (Lee et al., 2024) to convert a pre-trained score model into a flow model (as described in Appendix D.1), the network is wrapped in a velocity-prediction head. This wrapper adapts the model's output to predict a velocity field, making it directly compatible with

Table 8: Comparisons of various methods in unconditional CIFAR-10 image generation. In the column "#Batch", the number of training batches of each baseline method is in black, while that of OAT-FM is in purple. The unit "K" means 1,000 batches (each batch contains 128 samples). All results are obtained using the Euler solver.

| Method | #Batch | NFE↓ | FID↓ |
|---|---|---|---|
| FM (Lipman et al., 2023) | 400K | 100 (Euler) | 4.600 |
| FM + OAT-FM | +1K | 100 (Euler) | **3.917** |
| I-CFM (Tong et al., 2024) | 400K | 100 (Euler) | 4.404 |
| I-CFM + OAT-FM | +1K | 100 (Euler) | **3.784** |
| OT-CFM (Tong et al., 2024) | 400K | 100 (Euler) | 4.492 |
| OT-CFM + OAT-FM | +1K | 100 (Euler) | **3.734** |

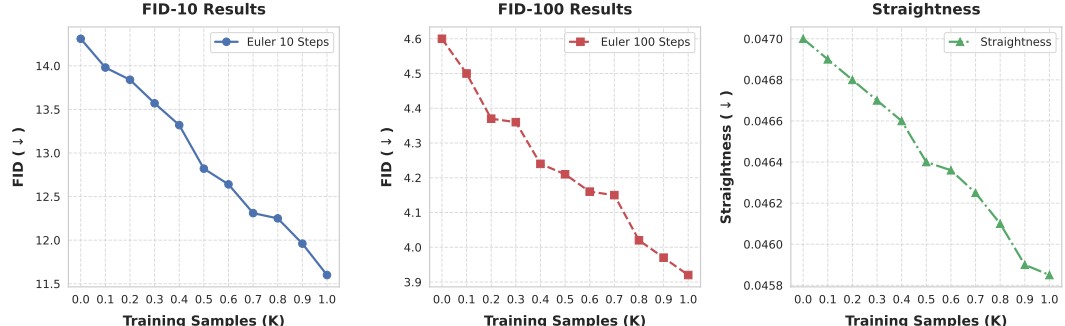

Figure 5: Stability analysis of OAT-FM fine-tuning on CIFAR-10. Starting from a pre-trained FM model (400K batches), we refine the model using OAT-FM for an additional 1K training samples, recording metrics every 0.1K samples. We track generation quality (FID with 10 and 100 Euler steps) and flow straightness.

our flow matching objective. Using OAT-FM, it is trained for an additional 12K iterations in EMA strategy, with a learning rate warmup over the first 1K iterations from 3e-5 to 3e-4.

**Evaluation**. In the inference phase, We employ the adaptive Dopri5 solver for FM, I-CFM, and OT-CFM, and the Heun solver with 35 steps for EDM. For each model, we report image quality measured by the Fréchet Inception Distance (FID) (Heusel et al., 2017), along with the number of training iterations (#Iter.) and the number of function evaluations (NFE) during inference.

### D.3.2    MORE EXPERIMENT RESULTS

**Additional Results wth Euler Solvers**. In addition, we also present the results of FM, I-CFM, and OT-CFM using the Euler solver with 100 steps in Table 8.

**Stability Analysis**. To investigate the stability of our method and verify that the OAT-FM refinement does not lead to deterioration of the transport map or distribution drift over time, we conducted a fine-grained quantitative analysis on the CIFAR-10 dataset. We initialize our training from the pre-trained FM model (Lipman et al., 2023) and fine-tune it with OAT-FM on an additional 1K training samples, recording performance metrics per 0.1K samples. As shown in Figure 5, we track three key metrics: generation quality using an Euler solver with 10 steps (FID-10) and 100 steps (FID-100), and the flow straightness score (Liu, 2022; Lee et al., 2024).[5] We observe a consistent, monotonic improvement across all three metrics as the number of OAT-FM training samples increases. This validates the stability of the proposed two-phase training paradigm.

---

[5]The straightness score is computed with 100 integration steps by measuring the mean squared error between the actual velocities along the ODE trajectory and the constant velocity from the initial to the final state, with lower values indicating straighter flows.

Table 9: Sensitivity analysis of OAT-FM to Phase 1 model quality on CIFAR-10. We evaluate the efficacy of OAT-FM fine-tuning (for only 1K batches) applied to FM, I-CFM, and OT-CFM checkpoints trained for varying durations (100K to 400K batches).

| Method | 100K Batches | | 200K Batches | | 300K Batches | | 400K Batches | |
|---|---|---|---|---|---|---|---|---|
| | NFE↓ | FID↓ | NFE↓ | FID↓ | NFE↓ | FID↓ | NFE↓ | FID↓ |
| FM | 140 | 6.11 | 140 | 4.26 | 143 | 3.88 | 147 | 3.71 |
| + OAT-FM (1K) | **135** | **5.60** | **132** | **3.96** | **134** | **3.63** | **135** | **3.54** |
| I-CFM | 140 | 5.97 | 140 | 4.13 | 140 | 3.81 | 149 | 3.67 |
| + OAT-FM (1K) | **131** | **5.60** | **137** | **3.95** | **134** | **3.44** | **138** | **3.48** |
| OT-CFM | 138 | 6.23 | 133 | 4.40 | 134 | 3.93 | 132 | 3.64 |
| + OAT-FM (1K) | **128** | **6.02** | **126** | **4.18** | **128** | **3.71** | **126** | **3.46** |

**Sensitivity to Phase 1 pre-trained model**. As presented in Table 9, we evaluated the efficacy of OAT-FM when applied to pre-trained models at different stages of convergence. Specifically, we utilized checkpoints from FM, I-CFM, and OT-CFM trained for 100K, 200K, 300K, and 400K batches on CIFAR-10. OAT-FM (applied for only 1K batches) consistently improves the generation quality (FID) and reduces the Number of Function Evaluations (NFE) across all initialization points. Notably, even for less-converged models (e.g., FM at 100K batches), OAT-FM successfully reduces the FID from 6.11 to 5.60, demonstrating that our method does not require a near-perfect velocity field to yield benefits. This indicates that OAT-FM is robust to the quality of the initial velocity estimates and avoids catastrophic performance degradation even when the Phase 1 pre-trained model is suboptimal. Furthermore, our method demonstrates significant training efficiency. For instance, the FM model trained for 300K batches with OAT-FM refinement matches the performance of the stronger OT-CFM baseline trained for the whole 400K batches (FID 3.63 vs. 3.64), and the I-CFM model at 300K with refinement (FID 3.44) outperforms the fully converged OT-CFM baseline (FID 3.44 vs. 3.64).

### D.4 IMAGENET 256×256

#### D.4.1 BASIC SETTINGS

**Dataset**. We extend our evaluation to class-conditional generation on the ImageNet 256×256 benchmark (Deng et al., 2009). This dataset, a standard for large-scale image generation, consists of approximately 1.28 million training images, categorized into 1,000 classes.

**Training**. Our generative model for ImageNet is a Scalable Interpolant Transformer (SiT) (Ma et al., 2024), which utilizes the Diffusion Transformer (DiT) (Peebles & Xie, 2023) backbone. We downloaded the checkpoint from `https://github.com/willisma/SiT`. The model operates in the latent space of a pre-trained variational autoencoder (VAE) (Kingma & Welling, 2013). The network architecture is the XL/2 version of SiT. The interpolant framework allows for flexible choices of path-type and model prediction targets, with linear path and velocity prediction being the default configuration. We use the AdamW optimizer with a learning rate of $1 \times 10^{-4}$ and no weight decay. The target network is updated via an EMA strategy of the online model's weights with a decay of 0.9999. For the OAT-FM objective, we set the balancing hyperparameter $\alpha$ to 0.80. When applying Algorithm 1, to prevent cross-class interference within mini-batches, our sampling strategy assigns a unique class to each GPU for every training iteration. This ensures that each local batch consists solely of images from a single class. To incorporate Classifier-Free Guidance (CFG), we follow the training protocol of SiT-XL and randomly drop class labels with a probability of 0.1. This procedure partitions each mini-batch into two subsets: a conditional (labeled) group and an unconditional (unlabeled) group. Our method computes the OAT plan and performs the corresponding sample pairing independently within each subset. Finally, the paired samples from both groups are combined to compute the training loss. The pre-trained SiT-XL is trained by OAT-FM with additional 48K iterations or 5 epochs. Training performance metrics (including memory and training time) are detailed in Table 10.

**Evaluations**. We employ both ODE and SDE samplers for evaluations. For ODE-based sampling, we utilize the adaptive step-size Dopri5 solver (Dormand & Prince, 1980), configured with an ab-

Table 10: Training efficiency and resource consumption of SiT-XL with and without OAT-FM across different GPU configurations.

| Model | GPU | Batchsize | Peak Allocated / Reserved Memory | Wall-clock Training Time | |
|---|---|---|---|---|---|
| | | | | Speed | 5 epochs |
| SiT-XL | A6000 × 8 | 128 × 8 | 35.57 GB / 44.61 GB | 276 samples/s | ∼6.1 hours |
| + OAT-FM | A6000 × 8 | 128 × 8 | 35.61 GB / 44.62 GB | 225 samples/s | ∼7.5 hours |
| SiT-XL | A100 × 8 | 128 × 8 | 35.57 GB / 44.49 GB | 584 samples/s | ∼2.9 hours |
| + OAT-FM | A100 × 8 | 128 × 8 | 35.61 GB / 44.49 GB | 440 samples/s | ∼3.8 hours |
| SiT-XL | A100 × 8 | 256 × 8 | 55.67 GB / 72.61 GB | 614 samples/s | ∼2.7 hours |
| + OAT-FM | A100 × 8 | 256 × 8 | 55.91 GB / 72.83 GB | 452 samples/s | ∼3.7 hours |

Table 11: Comparisons of SiT-XL with and without OAT-FM across different CFG scales. In the column "#Epochs", the number of training epochs of each baseline method is in black, while that of OAT-FM is in purple.

| Method | #Epochs | FID↓ | sFID↓ | IS↑ | P↑ | R↑ |
|---|---|---|---|---|---|---|
| SiT-XL$_{\text{CFG=1.0, Sampler=ODE}}$ | 1,400 | 9.40 | **6.39** | 125.2 | **0.67** | **0.67** |
| SiT-XL$_{\text{CFG=1.0, Sampler=ODE}}$ + OAT-FM | +5 | **9.18** | 6.42 | **128.5** | **0.67** | **0.67** |
| SiT-XL$_{\text{CFG=1.5, Sampler=ODE}}$ | 1,400 | 2.11 | **4.62** | 256.0 | **0.81** | **0.61** |
| SiT-XL$_{\text{CFG=1.5, Sampler=ODE}}$ + OAT-FM | +5 | **2.05** | 4.62 | **259.4** | 0.80 | **0.61** |
| SiT-XL$_{\text{CFG=2.0, Sampler=ODE}}$ | 1,400 | 3.89 | 5.02 | 342.5 | **0.87** | **0.54** |
| SiT-XL$_{\text{CFG=2.0, Sampler=ODE}}$ + OAT-FM | +5 | **3.74** | **4.84** | **346.6** | 0.86 | **0.54** |
| SiT-XL$_{\text{CFG=2.5, Sampler=ODE}}$ | 1,400 | 6.91 | 6.42 | 391.5 | **0.89** | 0.47 |
| SiT-XL$_{\text{CFG=2.5, Sampler=ODE}}$ + OAT-FM | +5 | **6.57** | **5.98** | **394.8** | **0.89** | **0.49** |
| SiT-XL$_{\text{CFG=3.0, Sampler=ODE}}$ | 1,400 | 9.31 | 8.10 | 419.2 | **0.90** | 0.41 |
| SiT-XL$_{\text{CFG=3.0, Sampler=ODE}}$ + OAT-FM | +5 | **8.87** | **7.41** | **421.9** | **0.90** | **0.44** |
| SiT-XL$_{\text{CFG=3.5, Sampler=ODE}}$ | 1,400 | 11.14 | 9.80 | 435.8 | **0.91** | 0.37 |
| SiT-XL$_{\text{CFG=3.5, Sampler=ODE}}$ + OAT-FM | +5 | **10.55** | **8.84** | **437.5** | 0.90 | **0.39** |
| SiT-XL$_{\text{CFG=4.0, Sampler=ODE}}$ | 1,400 | 12.50 | 11.30 | 444.8 | **0.91** | 0.35 |
| SiT-XL$_{\text{CFG=4.0, Sampler=ODE}}$ + OAT-FM | +5 | **11.72** | **10.07** | **449.1** | 0.90 | **0.37** |

solute tolerance of $1 \times 10^{-6}$ and a relative tolerance of $1 \times 10^{-3}$. For SDE-based sampling, we use a fixed-step Euler-Maruyama solver with 250 steps. In both settings, we leverage classifier-free guidance to improve sample quality.

### D.4.2 MORE EXPERIMENT RESULTS

The results achieved under different CFG scales are shown in Table 11. In addition, we generate images of different classes by the original SiT-XL and that refined by OAT-FM, respectively. For each example, we start from the same noise point to generate images by the two models. Following existing methods (Ma et al., 2024; Peebles & Xie, 2023), we set the CFG scale to be 4.0 for good visual effects. Some typical results are shown in Figures 6 and 8, demonstrating that the refinement achieved by OAT-FM indeed leads to better image quality.

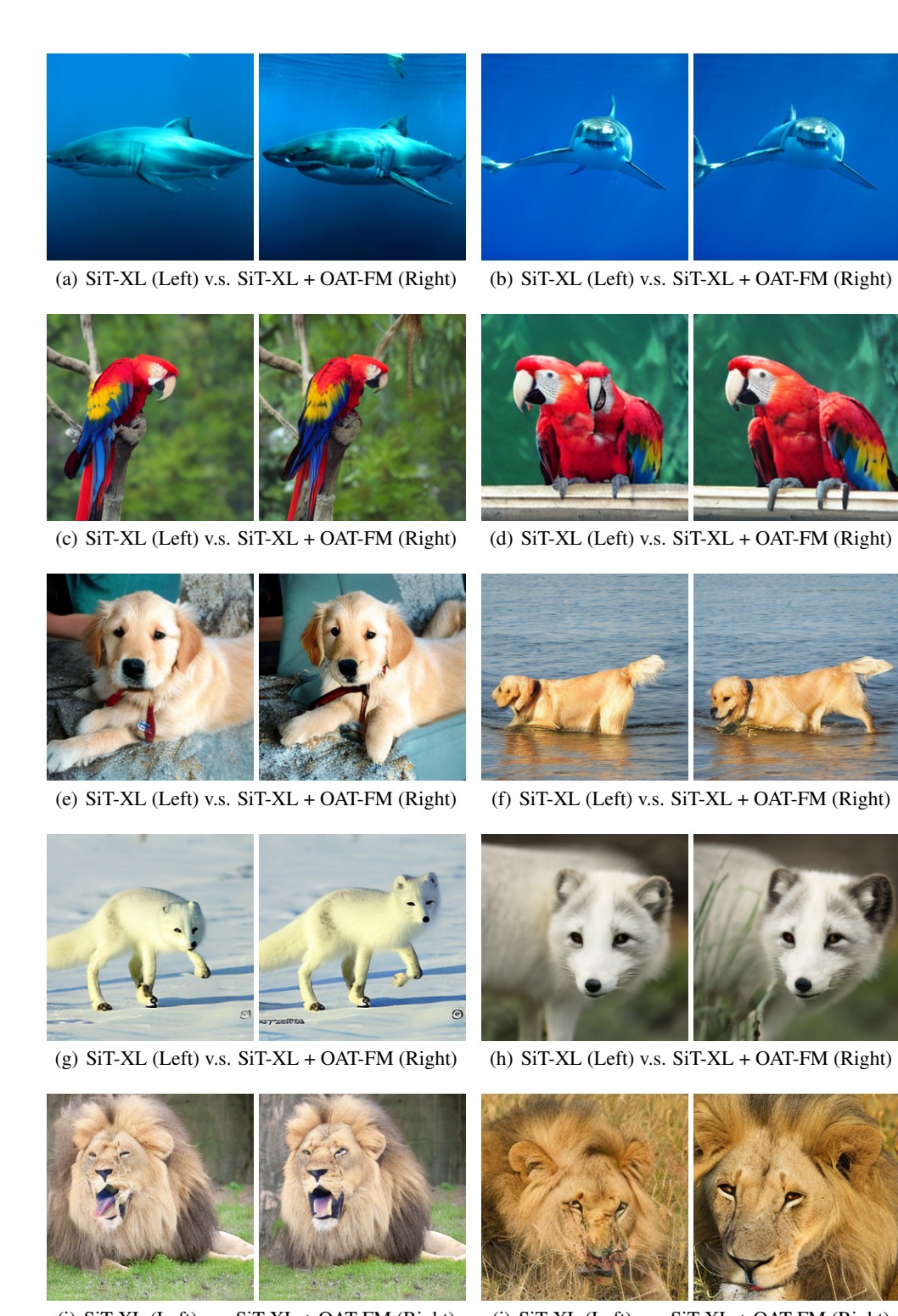

(a) SiT-XL (Left) v.s. SiT-XL + OAT-FM (Right)   (b) SiT-XL (Left) v.s. SiT-XL + OAT-FM (Right)

(c) SiT-XL (Left) v.s. SiT-XL + OAT-FM (Right)   (d) SiT-XL (Left) v.s. SiT-XL + OAT-FM (Right)

(e) SiT-XL (Left) v.s. SiT-XL + OAT-FM (Right)   (f) SiT-XL (Left) v.s. SiT-XL + OAT-FM (Right)

(g) SiT-XL (Left) v.s. SiT-XL + OAT-FM (Right)   (h) SiT-XL (Left) v.s. SiT-XL + OAT-FM (Right)

(i) SiT-XL (Left) v.s. SiT-XL + OAT-FM (Right)   (j) SiT-XL (Left) v.s. SiT-XL + OAT-FM (Right)

Figure 6: Some generation results achieved by the two methods.

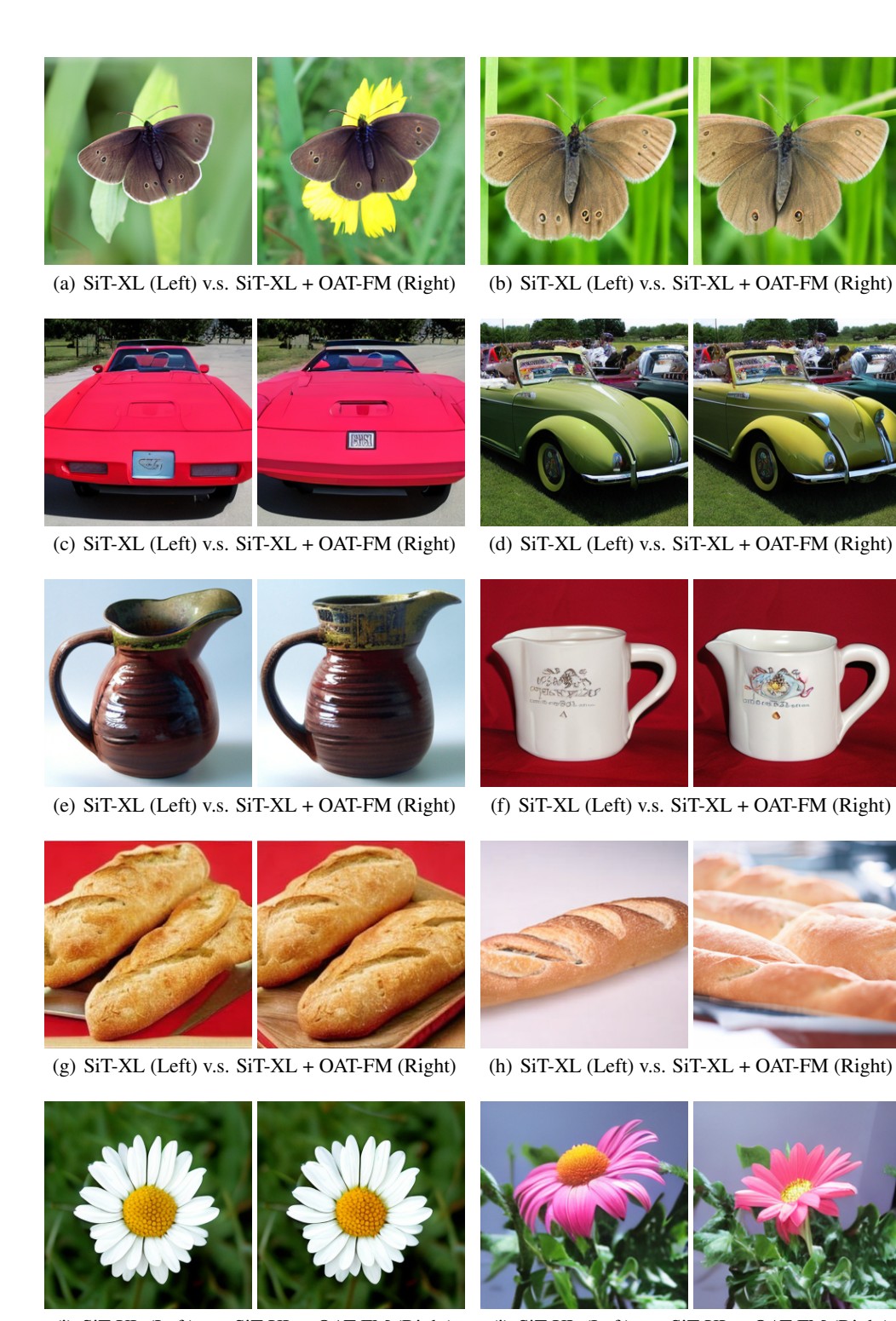

(a) SiT-XL (Left) v.s. SiT-XL + OAT-FM (Right)      (b) SiT-XL (Left) v.s. SiT-XL + OAT-FM (Right)

(c) SiT-XL (Left) v.s. SiT-XL + OAT-FM (Right)      (d) SiT-XL (Left) v.s. SiT-XL + OAT-FM (Right)

(e) SiT-XL (Left) v.s. SiT-XL + OAT-FM (Right)      (f) SiT-XL (Left) v.s. SiT-XL + OAT-FM (Right)

(g) SiT-XL (Left) v.s. SiT-XL + OAT-FM (Right)      (h) SiT-XL (Left) v.s. SiT-XL + OAT-FM (Right)

(i) SiT-XL (Left) v.s. SiT-XL + OAT-FM (Right)      (j) SiT-XL (Left) v.s. SiT-XL + OAT-FM (Right)

Figure 7: Some generation results achieved by the two methods.

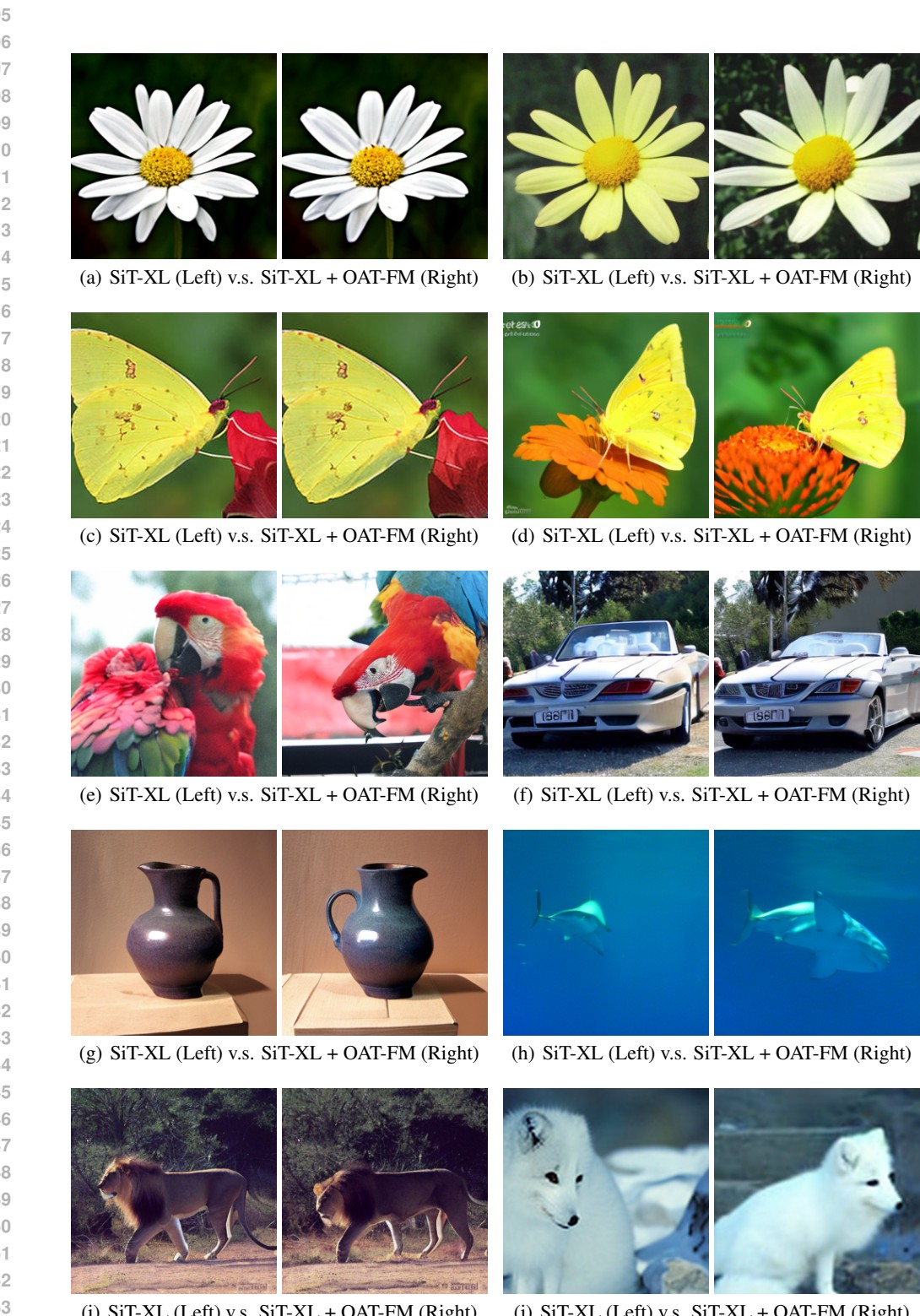

(a) SiT-XL (Left) v.s. SiT-XL + OAT-FM (Right)     (b) SiT-XL (Left) v.s. SiT-XL + OAT-FM (Right)

(c) SiT-XL (Left) v.s. SiT-XL + OAT-FM (Right)     (d) SiT-XL (Left) v.s. SiT-XL + OAT-FM (Right)

(e) SiT-XL (Left) v.s. SiT-XL + OAT-FM (Right)     (f) SiT-XL (Left) v.s. SiT-XL + OAT-FM (Right)

(g) SiT-XL (Left) v.s. SiT-XL + OAT-FM (Right)     (h) SiT-XL (Left) v.s. SiT-XL + OAT-FM (Right)

(i) SiT-XL (Left) v.s. SiT-XL + OAT-FM (Right)     (j) SiT-XL (Left) v.s. SiT-XL + OAT-FM (Right)

Figure 8: Some generation results achieved by the two methods.

