# OpenReview forum: "OAT-FM: Optimal Acceleration Transport for Improved Flow Matching"
_ICLR.cc/2026/Conference — Submitted to ICLR 2026_

### Official Review · Reviewer_BrGa · 2025-10-24

**Soundness:** 3
**Presentation:** 3
**Contribution:** 3
**Rating:** 6
**Confidence:** 4

**Summary:**

This paper introduces OAT-FM, a new Flow Matching (FM) method inspired by the theory of Optimal Acceleration Transport (OAT). Standard Optimal Transport (OT) based FM methods aim to straighten flows by minimizing kinetic energy (velocity), which is a sufficient but not necessary condition for straightness. The authors argue that a more fundamental condition for straightness is minimizing acceleration.

The proposed OAT-FM lifts the transport problem from the sample space $\mathcal{X}$ to the product space of sample and velocity, $\mathcal{X} \times \mathcal{V}$. The objective is to find a coupling that minimizes the total squared acceleration, which the authors show is equivalent to a necessary and sufficient condition for flow straightness.

Because this new objective requires reliable velocity estimates at the endpoints (noise and data), which are not available at the start of training, OAT-FM is proposed as a **two-phase paradigm**. Phase 1 involves training any standard flow/diffusion model to obtain "relatively reliable" velocity information. Phase 2 then fine-tunes this pretrained model using the OAT-FM objective, which refines the flow trajectories by minimizing acceleration. The authors provide an efficient algorithm that reduces the complex OAT coupling in the product space to a standard OT problem in the sample space, giving it the same asymptotic complexity as OT-CFM.

Experiments on 2D benchmarks, CIFAR-10, and ImageNet $256^2$ show that this OAT-FM fine-tuning phase consistently improves the performance (e.g., FID) of various strong pretrained models (including EDM and SiT-XL) with very little computational overhead.

**Strengths:**

1.  **Novel Theoretical Framework:** The paper successfully connects Flow Matching with the theory of Optimal Acceleration Transport (OAT), providing a new, physically-grounded objective for straightening generative flows by minimizing acceleration.
2.  **Practical & Efficient Refinement:** The proposed two-phase paradigm is highly practical. It provides a way to improve existing, strong generative models with a very small additional compute budget. The empirical results (e.g., improving SiT-XL on ImageNet with only +5 epochs) are compelling.
3.  **Strong Empirical Validation:** The method is shown to consistently improve a wide array of baseline models (FM, I-CFM, OT-CFM, EDM, SiT-XL) across diverse tasks (2D OT, CIFAR-10, ImageNet 256).
4.  **Computationally Feasible:** The authors provide a crucial simplification (Section 2.3) that reduces the complex OAT coupling problem in $\mathcal{X} \times \mathcal{V}$ to a standard OT problem in $\mathcal{X}$, making its complexity on par with existing OT-CFM methods.

**Weaknesses:**

**Computational Scaling:** While the asymptotic complexity is the same as OT-CFM, the lower-level problem still requires solving a minibatch OT problem, which scales quadratically with the batch size $B$ (e.g., $\mathcal{O}(B^2 \log B)$ with Sinkhorn), as noted in the limitations. This could be a practical bottleneck for scaling to very large batch sizes.

**Questions:**

1. The cost function $l_{\mathcal{A}}$ (Eq 6) has a hyperparameter $\alpha$. The proof of Theorem 3 suggests $\alpha=2/3$ maximizes the theoretical bound, but the text also mentions $\alpha=12/13$ when relating Eq 6 to Eq 5. What value of $\alpha$ was used in the experiments, and how sensitive is the final performance (e.g., FID) to this choice?
2.  There seems to be a slight discrepancy in the target network updates mentioned in the appendix. Appendix D.2 (2D task) mentions a "hard copy... every 500 batches," while Appendix D.3 (CIFAR-10) mentions an "EMA... with a decay of 0.9999." Which strategy was used for the ImageNet (SiT-XL) experiments? Is the method sensitive to this choice (hard vs. soft update)?

---

> ### Author Response · Authors · 2025-11-22
> **Response to Reviewer BrGa**
>
> Thanks for your appreciation of our work. We have revised our paper, adding more analytical content and experiments to address your concerns. Below, we answer your questions one by one.
>
> **Q1: Computational Scalability**
>
> **A1:** Minibatch-OT indeed suffers from the scalability issue when the batch size is very large. This problem is common for all primal-formed OT algorithms, not specific to our OAT-FM. Below, we would like to highlight our contributions in computation.
>
> 1) **Make OAT from inapplicable to practical.** Given a batch of $B$ samples and $B$ velocities, the original complexity of OAT is at least $\mathcal{O}(B^3)$, which is too high to practical applications. We leverage a decomposable formulation (Eq.(9)) to reduce the complexity to $\mathcal{O}(B^2\log B)$ (equivalent to that of standard OT), making an intractable method practical.
>
> 2) **Demonstrate the empirical efficiency of OAT-based FM.** Most existing OT-based FM methods are trained and tested on simple tasks (e.g., CIFAR-10). In our work, we scale OAT-FM to ImageNet-level image generation. To further resolve your concern, we provide detailed training costs of OAT-FM in the following table (**Table 10 in the revised paper**), demonstrating that the quadratic scaling of the mini-batch OT solver introduces modest overhead in the context of FM.
>
> - **Competitive Speed & Negligible Memory:** OAT-FM maintains 73%-80% speed of the baseline (SiT-XL) while keeping memory consumption almost identical. In practice, a global batch size of 1024 or 2048 is sufficient for ImageNet and most generation tasks, which our method handles effortlessly.
>
> - **Short Total Training Time:** Although OAT-FM introduces additional computations per step, it is highly data-efficient, requiring only 5 epochs for the phase-2 training, whose total training duration is $<$ 4 hours on 8 A100 GPUs.
>
> ||GPU|Batchsize|Peak Allocated/Reserved Memory (GB)|Speed (samples/s)|Time (Hours per 5 epochs)|
> |-|-|-|-|-|-|
> |SiT-XL|A6000×8|128×8|35.57/44.61| 276 |6.1|
> |+OAT-FM|A6000×8|128×8| 35.61/44.62|225|7.5|
> |SiT-XL|A100×8|128×8|35.57/44.49|584|2.9|
> |+OAT-FM|A100×8|128×8|35.61/44.49|440|3.8|
> |SiT-XL|A100×8|256×8|55.67/72.61|614|2.7|
> |+OAT-FM|A100×8|256×8|55.91/72.83|452|3.7|
>
> To further improve per-step computation speed, we are actively investigating the dual formulation of OAT, as Optimal Flow Matching (OFM) [1] does. We leave this as our future work.
>
> **Q2. The setting of hyperparameter $\alpha$**
>
> **A2:** The parameter $\alpha$ balances velocity alignment against the acceleration penalty, as shown in Lines 211-214. Empirically, we observe robust performance within the range $\alpha \in [0.7, 0.8]$. In our experiments, we set $\alpha=0.7$ for toy tasks, $0.75$ for CIFAR-10, and $0.80$ for ImageNet. In the revised paper, we have added the settings in the Appendix.
>
> Note that setting $\alpha=12/13$ in Eq.(6) aims to connect our loss to the cost of OAT in Eq.(5), which emphasizes the motivation of our loss design. In Theorem 3, setting $\alpha=2/3$ leads to an OAT-based lower bound of our loss, which helps verify the rationality of our loss. These theoretical values do not necessarily correspond to the optimal settings for stable and effective training in practice.
>
> **Q3.The rationality of implementation (e.g., Hard copy v.s. EMA update)**
>
> **A3:** For low-dimensional toy tasks (2D), a straightforward 'hard copy' strategy is sufficient to achieve excellent performance due to the simplicity of the task, as shown in Table 2. For high-dimensional generative tasks, EMA update is a common strategy [2,3], and we follow their settings for a fair comparison.
>
> [1] Kornilov, Nikita, et al. "Optimal flow matching: Learning straight trajectories in just one step." Advances in Neural Information Processing Systems 37 (2024): 104180-104204.
>
> [2] Song, Yang, et al. "Consistency models." (2023).
>
> [3] Tong, Alexander, et al. "Improving and generalizing flow-based generative models with minibatch optimal transport." arXiv preprint arXiv:2302.00482 (2023).
>
> In summary, we hope the above response resolves your concerns and makes you more confident in raising your score in the following discussion and decision phases. Thanks in advance for your consideration.

---

> > ### Comment · Reviewer_BrGa · 2025-11-27
> >
> > Thanks for responding. The authors have addressed most of my concerns. I will keep my score.

---

> ### Author Response · Authors · 2025-12-03
> **Thank You for Your Feedback**
>
> Thank you for your feedback and continued support. We are pleased that our clarifications have addressed your concerns.

---

### Official Review · Reviewer_byYL · 2025-10-27

**Soundness:** 2
**Presentation:** 2
**Contribution:** 3
**Rating:** 4
**Confidence:** 3

**Summary:**

The authors propose a novel fine-tuning method for flow models inspired by Optimal Acceleration Transport (OAT). They first introduce the theoretical framework of OAT and derive a practical optimization objective based on minimizing acceleration. The effectiveness of the method is then validated on both low- and high-dimensional datasets, including ablation studies that demonstrate the importance of the proposed loss components.

**Strengths:**

- The authors explore a promising direction in optimal transport by advancing from first-order to second-order formulations.
- The proposed method maintains computational complexity comparable to first-order approaches and does not require explicit modeling of acceleration fields.
- The comparison to prior second-order methods is clear and highlights the key differences effectively.

**Weaknesses:**

**Theory**. Reducing acceleration may locally straighten trajectories, but the paper does not explicitly discuss how this affects preservation of the transport map from $\mu_0$ to $\mu_1$. Without enforcing a Vlasov constraint, it is unclear whether mass conservation and global flow validity are maintained, i.e., if the method (final loss 7) is theoretically valid.

**Clarity.** The previous concern is also closely related to the overall insufficient clarity of the paper. Although the authors seem to discuss the main required theory and derivations, they are still hard to parse and understand their relations, even for a reader experienced in optimal transport. Furthermore, the paper’s theory relies on a recent paper (Brigati et al., 2025) which, to my knowledge, is still not reviewed or published.

**Practice**.
- Based on the theoretical concern above, it is essential to evaluate how the refinement behaves as fine-tuning iterations increase, for example by tracking changes in quality metrics and trajectory straightness, similar to the analyses shown in Figures 8b and 9 of [1]. This could reveal whether the transport map remains valid or gradually deteriorates under more aggressive application of OAT-FM. A plot of FID as a function of the training (fine-tunning) epoch should be provided.
- The proposed methods do not show substantial performance improvements, particularly in higher-dimensional settings (Section 4.2). This is an important weakness given the heuristical nature of the method. I am not convinced that the improvements are statistically significant (experiments in 4.1 are toys and can be considered only for illustration purposes).
- The rationale for selecting only a specific subset of baseline methods is not explained. For example, although OFM [2] is discussed in the paper, it is not included in the empirical comparison.

[1] Liu, Xingchao, Chengyue Gong, and Qiang Liu. "Flow straight and fast: Learning to generate and transfer data with rectified flow."

[2] Kornilov, Nikita, et al. "Optimal flow matching: Learning straight trajectories in just one step."

**Typos, etc.**
- In line 135 a dot is missing “...respectively Note…”.
- In line 480 a wrong capitailisation “...so Currently…”.
- Figure 1a does not clearly convey the main idea of the method. A more intuitive and visually expressive illustration would help readers better understand the key concepts.

**Questions:**

- In line 906 it is not clear why the highlighted term is irrelevant w.r.t. $\pi_x$. If I am not mistaken, this terms contains a cross product $v_1^{T}v_0$ which directly depends on the coupling for x because $v_0$ and $v_1$ are determined using them.
- For other comments and questions, see the weaknesses section.

---

> ### Author Response · Authors · 2025-11-22
> **Response to Reviewer byYL (Part 1)**
>
> Thanks for your constructive comments. We have revised our paper according to your comments, fixing typos and adding more explanations and experiments. For Figure 1a, we add more notations and explanations to highlight the principle of our method. Below, we answer your questions one by one.
>
> **Q1 (Theory & Clarity): The theoretical properties of the method, e.g., the mass conservation and the validity of learned flow**
>
> **A1:** Our final method is theoretically valid due to the properties of OAT theory and conditional flow matching.
>
> 1) **Mass conservation and Vlasov enforcement.** As shown in Theorem 3, the loss function of our method is lower bounded by the OAT problem in the static Kantorovich-formed definition (Definition 2 in the paper). Chen et al. (2018) and Benamou et al. (2019) have shown that this static definition is equivalent to the dynamic one (Definition 1 in the paper), so that the constraint of the Vlasov equation is held implicitly. The Vlasov equation is also known as the second-order continuity equation; therefore, the total mass is conserved by its generic property (Chen et al., 2018; Benamou et al., 2019; Brigati et al., 2025).
>
> 2) **Flow matching guarantees.** OAT-FM actually achieves the OT-CFM in the product space of sample and velocity, whose objective is acceleration minimization (i.e., pursuing a smoothed velocity field). Standard CFM theory guarantees that the aggregated vector field successfully generates a flow transporting $\mu_0$ to $\mu_1$ in the product space.
>
> **Q2 (Theory & Clarity): The solidness of OAT theory itself, and its connection to the proposed method**
>
> **A2:** Although the paper (Brigati et al., 2025) is on arXiv, the OAT formulation in this paper has been verified by many publications. In particular, the Kantorovich formulation of OAT was first proposed and discussed by Chen et al. (2018, Eq. 10) and Benamou et al. (2019, Eq. 4.14). The reason for citing (Brigati et al., 2025) is that the OAT formulation used in our work is the same as that in this paper. On page 4, we have cited more published papers and added a footnote to explain the relationship among them.
>
> The connection between the OAT problem and our OAT-FM method is analogous to that between the OT problem and OT-CFM. In particular, it has been well known that OT-CFM learns a flow that achieves optimal transport in the sample space, straightening flow trajectories by pursuing constant velocity. Our OAT-FM learns a flow to achieve optimal acceleration transport in the product space of sample and velocity, straightening flow trajectories by minimizing acceleration (i.e., a smooth velocity field). We have added the above analysis in the revised paper (Lines 226-232).
>
> **Q3 (Practice): Analyze the refinement behaviors as fine-tuning iterations increase.**
>
> **A3:** Following your suggestion, we conducted a fine-grained quantitative analysis on the CIFAR-10 dataset, as shown in **Figure 5 and Lines 1202-1210** in the revised paper.
>
> We initialize our training from the pre-trained FM model and fine-tune it with OAT-FM using an additional 1K training samples, recording performance metrics every 0.1K samples. As shown in **Figure 5**, we track three key metrics: generation quality using an Euler solver with 10 steps (FID-10) and 100 steps (FID-100), and the flow straightness score. We observe a consistent, monotonic improvement across all three metrics as the number of OAT-FM training samples increases. **This validates the stability of the proposed two-phase training paradigm --- improvement achieved by our refinement is consistent.**
>
> **Q4: The highlighted term in the derivation of Eq.(9).**
>
> **A4:** Sorry for this confusion. We use "irrelevant to $\pi_x$" to emphasize that this term is purely determined by the parametric velocity. In the revised paper, we have rewritten the derivation of Eq. (9) using new notation and explained our derivation in **Lines 948-990**. In particular, we remove the confusing expression "irrelevant to $\pi_x$" and denote the highlighted term as $C$. We ignore it in our implementation to stabilize the training process --- this term is purely determined by the parametric velocity, which may change significantly during training.

---

> ### Author Response · Authors · 2025-11-22
> **Response to Reviewer byYL (Part 2)**
>
> **Q5 (Practice): The significance of improvement in high-dimensional tasks.**
>
> **A5:** We respectively disagree with the comment that our improvement is not significant in high-dimensional tasks. As shown in Figure 3, when cfg=4.0, a common setting for generating high-resolution images, our method achieves significant improvements in FID and sFID (gains of around 1). **Table 11 in Appendix D.4.2** provides a more detailed numerical comparison. In addition, for the CIFAR-10 experimental results in Table 3, the improvement from FM to FM+OAT-FM (3.71->3.54) is more significant than that from FM to OT-CFM (3.71->3.64).
>
> To further resolve your concern, in Table 9 of the revised paper, we evaluate the efficacy of OAT-FM fine-tuning (applied to only 1K batches) on FM, I-CFM, and OT-CFM checkpoints trained for varying durations (100K to 400K batches). The results demonstrate that our method consistently achieves significant improvements, even when the phase-1 models are not well-trained.
>
> ||100K NFE|100K FID|200K NFE|200K FID|300K NFE|300K FID|400K NFE|400K FID|
> |-|-|-|-|-|-|-|-|-|
> |FM|140|6.11|140|4.26|143|3.88|147|3.71|
> |+OAT-FM (1K)|**135**|**5.60**|**132**|**3.96**|**134**|**3.63**|**135**|**3.54**|
> |I-CFM|140|5.97|140|4.13|140|3.81|149|3.67|
> |+OAT-FM (1K)|**131**|**5.60**|**137**|**3.95**|**134**|**3.44**|**138**|**3.48**|
> |OT-CFM|138|6.23|133|4.40|134|3.93|132|3.64|
> |+OAT-FM (1K)|**128**|**6.02**|**126**|**4.18**|**128**|**3.71**|**126**|**3.46**|
>
> In Figure 4, in the revised paper, we present enlarged visual results for SiT-XL and SiT-XL+OAT-FM for better comparison and analyze their differences (**Lines 512-521**). In particular, when starting from the same noise, in some cases (e.g., the "Bottle" and "Parrot" shown in Figure 4), applying OAT-FM improves image details, maintains the main semantic and spatial content generated by SiT-XL, and suppresses hallucination. In the other cases (e.g., "Lion" and "Car" shown in Figure 4), applying OAT-FM results in the model generating entirely different images compared to those produced by SiT-XL. This phenomenon indicates that for flow trajectories whose endpoints closely follow the data distribution, OAT-FM simply straightens them while preserving their endpoints. For flow trajectories whose endpoints are undesired (e.g., in the "Lion" generated by SiT-XL), OAT-FM significantly changes their directions, leading to new endpoints that may better fit the data distribution.
>
> **Q6 (Practice): Add a comparison with OFM**
>
> **A6:** In the original paper, we did not consider OFM because of two reasons:
>
> 1) In low-dimensional tasks, we leverage the codebase of OT-CFM, in which all the methods use an MLP as their model. However, OFM is restricted to using the ICNN architecture. The authors of OFM claimed in their paper that the model fails to converge when replacing ICNN with MLP. As a result, comparing OFM with other methods is, to some extent, unfair --- it is not certain whether the performance difference is due to the learning method or the model architecture.
>
> 2) In high-dimensional tasks, OFM did not provide checkpoints for CIFAR-10 and ImageNet, and refactoring the code is non-trivial --- besides using ICNN, it requires a well-trained adversarial latent autoencoder (ALAE) for each specific data.
>
> To resolve your concern, in the low-dimensional benchmark, we compare OFM (using ICNN) with OAT-FM (using MLP) in a one-step generation setting. The data fitting (2-Wasserstein) results are in Table 7, pasted here for a quick review, showing the advantage of OAT-FM.
>
> ||$\mathcal{N}\to\text{8gs}$|$\text{8gs}\to\text{moons}$|$\mathcal{N}\to\text{moons}$|$\mathcal{N}\to\text{scurve}$|$\text{moons}\to\text{8gs}$|
> |-|-|-|-|-|-|
> |OFM|$0.71\pm0.27$|$0.22\pm0.02$|$0.22\pm0.01$|$1.99\pm0.25$|$0.46\pm0.11$|
> |FM|$21.50\pm0.07$|$4.05\pm0.01$|$8.08\pm0.05$|$79.01\pm1.18$|$13.48\pm0.15$|
> |+OAT-FM|$\mathbf{0.43\pm0.08}$|$\mathbf{0.14\pm0.02}$|$\mathbf{0.12\pm0.02}$|$\mathbf{1.41\pm0.23}$|$\mathbf{0.32\pm0.07}$|
> |I-CFM|$23.46\pm0.15$|$4.47\pm0.08$|$7.86\pm0.03$|$79.31\pm1.41$|$14.55\pm0.28$|
> |+OAT-FM|$\mathbf{0.42\pm0.10}$|$\mathbf{0.17\pm0.02}$|$\mathbf{0.12\pm0.01}$|$\mathbf{1.41\pm0.26}$|$\mathbf{0.51\pm0.13}$|
> |VP-CFM|$12.17\pm0.29$|$8.46\pm0.25$|$3.70\pm0.05$|$47.73\pm1.94$|$7.51\pm0.21$|
> |+OAT-FM|$\mathbf{0.45\pm0.11}$|$\mathbf{0.20\pm0.03}$|$\mathbf{0.12\pm0.01}$|$\mathbf{1.35\pm0.23}$|$\mathbf{0.36\pm0.07}$|
> |SB-CFM|$0.63\pm0.09$|$0.19\pm0.04$|$0.20\pm0.04$|$1.91\pm0.35$|$0.35\pm0.05$|
> |+OAT-FM|$\mathbf{0.42\pm0.10}$|$\mathbf{0.13\pm0.01}$|$\mathbf{0.10\pm0.01}$|$\mathbf{1.22\pm0.26}$|$\mathbf{0.33\pm0.07}$|
> |OT-CFM|$0.48\pm0.07$|$0.19\pm0.05$|$\mathbf{0.12\pm0.01}$|$1.64\pm0.22$|$0.35\pm0.08$|
> |+OAT-FM|$\mathbf{0.40\pm0.08}$|$\mathbf{0.13\pm0.00}$|$\mathbf{0.12\pm0.03}$|$\mathbf{1.23\pm0.19}$|$\mathbf{0.30\pm0.06}$|
>
> In summary, we hope the above response resolves your concerns and makes you confident in raising your score. Thanks in advance for your consideration.

---

### Official Review · Reviewer_FKVq · 2025-10-31

**Soundness:** 2
**Presentation:** 3
**Contribution:** 2
**Rating:** 4
**Confidence:** 4

**Summary:**

The paper under consideration introduces a framework, called OAT (Optimal Acceleration Transport) aimed at straightening the trajectories of a learned flow model with a novel OT-based technique. The core idea is to leverage the straightening problem in the joint phase space (coordinates times velocities) using a specific OT problem between probability distributions supported on product ($x \times v$) space. In the paper, the authors introduce the OAT problem itself; then adapt the OAT to the task of straightening flow-based models; and finally demonstrate the capabilities of the proposed methodology in a number of low and high-dimensional (e.g., deal with flow- and diffusion- based models on ImageNet 256x256) practical benchmarks.

**Strengths:**

- I found the narrative of the paper logical and easy-to-follow

- The adaptation and application of Optimal Acceleration Transport (which seems to be by itself a recently introduced theory) in the context of diffusion- and flow- based models is a novel and fresh (to my knowledge) idea in the community.

- The experiments demonstrate the competitiveness of the proposed framework in practice.

**Weaknesses:**

In a nutshell, the authors take the Optimal Acceleration Transport framework and adapt this framework to the problem of straightening flow-based models. My main concern and question regarding the paper is how this adaptation is achieved.

- At first, the optimized loss $\ell_{\mathcal{A}}$, eq. (6) is connected with OAT problem **rather indirectly**. To be honest, I do not understand the origin of the procedure described in lines 196-206, and subsequent OAT-FM problem formulation in eq. (7): we sample a trajectory from the original flow, consider an interpolation of the end points, then compute combination of $c_{\mathcal{A}}(z_0, z_t)$ and $c_{\mathcal{A}}(z_t, z_1)$ costs, and then solve an optimization problem with restrictions on top of this, eq. (7). While it is shown that this OAT-FM objective is indeed somehow related to OAT problem (Theorem 3), from my point this relation is rather weak and does not answer the question:  why the authors chose this particular formulation of the problem (eq. 7) rather than some other.

- Secondly, the implication, eq. (9) seems to be incorrect. I checked the reasoning in Appendix C.1 - it seems there is an error in the equations: the authors wrongly ignore velocities components in their deductions: it could not be done, even if you reduce the problem from $\pi$ to $\pi_x$ (lines 901 - 907), because the velocities are hardly connected with coordinates ($v = v(x)$) $\Rightarrow$ if we optimize a coupling between coordinates, we should treat terms like $\Vert v_1 - v_0 \Vert = \Vert v_1(x_1) - v_0 (x_0) \Vert$.

In light of this, I think that the proposed method (which seems to work competitively well in practice) requires better theoretical introduction and explanation.

**Questions:**

1. Theoretical question, line 162: We have Vlasov equation $\partial_t \mu + v \cdot \nabla_x \mu + \nabla_v \cdot (a \mu) = 0$. How $\nabla_x$ and $\nabla_v$ are defined? And what is $v$ in this equation (I think this is velocity. However, in equation 4, an auxiliary velocity field does not appear explicitly; it only appears implicitly as a sampled quantity from $\mu$.)?

2. From the practical results, e.g., Table 2, I found that compared to OT-CFM, OAT-FM typically leads to less straight trajectories, but better captures the marginal distribution. I found this strange, because OAT-FM (similar to ReFlow) operates with trained trajectories of an FM model, not with the original data, and, therefore, not expected to *improve* the marginals capturing. Could the authors comment on this?

---

> ### Author Response · Authors · 2025-11-22
> **Response to Reviewer FKVq**
>
> Thank you for your constructive comments. We have revised our paper based on your comments, adding more explanations and detailed derivations. Below, we answer your questions one by one.
>
> **Q1: Clarify the motivation of Eq.(6) and the relation between Eq.(7) and OAT problem.**
>
> **A1:** As explained in the revised paper (**Lines 226-232**), the connection between the OAT problem and our OAT-FM method is analogous to that between the OT problem and OT-CFM. In particular, it has been well known that OT-CFM learns a flow that achieves optimal transport in the sample space, straightening flow trajectories by pursuing constant velocity. **Our OAT-FM learns a flow to achieve optimal acceleration transport in the product space of sample and velocity, straightening flow trajectories by minimizing acceleration, which can be viewed as a second-order OT-CFM learning smoothed flows in the product space.** In the context, the design of our method is straightforward:
>
> 1) **The design of Eq.(6):** Eq.(6) introduces accleration terms (e.g., $\|v_\theta(x_t,t)-v_0\|$ and $\|v_1 - v_\theta(x_t,t)\|$, where the difference between velocities captures accleration) into the classic FM loss, weighted by $1-\alpha$, which leads to the cost function $c_A$ used in the Kantorovich form of OAT problem (Eq.(5)).
>
> 2) **The relation of Eq.(7) to OAT:** Similar to OT-CFM, we can formulate the OAT-FM problem as a bi-level optimization problem in Eq.(7). The expectation of the loss in Eq.(6) corresponds to the upper-level optimization problem (for flow matching). The distribution associated with the expectation (i.e., $\pi^*$) is determined by the lower-level problem, which is formulated as the OAT problem defined in Eq.(5) (for determining noise-data pairs).
>
> **Q2: The correctness of Eq.(9) and the corresponding acceleration strategy**
>
> **A2:** Sorry for this confusion. In Eq.(9), the $v$ means $v(x)$ indeed. We use this simplified notation to make the expression more concise and improve the equation's layout. We have revised it in Line 257.
>
> We also revised the derivation of Eq.(9) with new notations in **Lines 948-990**. We removed the confusing expression "irrelevant to $\pi_x$" used in the term $C$. Instead, we explain that we ignore this term to stabilize the training process --- this term is purely determined by the parametric velocity, which may change significantly during training.
>
> **Q3: Explain the Vlasov equation and its connection to Eq.(4)**
>
> **A3:** In the Vlasov equation $\partial_t\mu + v\cdot\nabla_x\mu + \nabla_v\cdot(a\mu)=0$, $\nabla_x$ and $\nabla_v$ mean "take the gradient with respect to $x$ and $v$", respectively.
>
> The dynamic form of the OAT problem (i.e., Eq.(4)) is an acceleration minimization problem, in which acceleration is a function of the sample $x$ and the velocity $v$ (i.e., $a(x,v,t)$). The objective in Eq. (4) involves integration over velocity. **In the dynamic form of the OAT problem, we optimize the acceleration field, rather than the velocity field. That is why you can only find the acceleration field in Eq.(4).**
>
> In practice, our OAT-FM models the acceleration field by differentiating the parametrized velocity field (i.e., $v_{\theta}$), then it learns a smoothed velocity field by solving an OAT problem.
>
> **Q4: The comparison for OT-CFM and OAT-FM on flow straightness.**
>
> **A4:** Firstly, we respectfully clarify that our OAT-FM is very different from Rectified Flow (ReFlow) in two points: 1) OAT-FM is trained on the original data, rather than generated noise-data pairs, and 2) OAT-FM matches noise with data by solving a mini-batch OAT problem (see Eq.(7)). As a result, OAT-FM avoids the distribution drifting issue of ReFlow and thus improves marginal capturing (lower 2-Wasserstein in Table 2).
>
> Secondly, as we explained in **A1**, OT-CFM straightens flow trajectories by maintaining constant velocity, while our OAT-FM straightens them by minimizing acceleration (i.e., pursuing a smoothed velocity field). **In other words, OAT-FM relaxes the straightness constraint but regularizes smoothness.** In complex high-dimensional manifolds (e.g., CIFAR-10, ImageNet), rigid straightness can be overly restrictive, and pursuing smoothness is a more reasonable inductive bias than rigid straightness, as it allows the flow to navigate the manifold geometry efficiently while maintaining high generation quality.
>
> For low-dimensional tasks in Table 2, OT-CFM trajectories are already near-perfectly straight due to the simplicity of the OT plan in the 2D space. Therefore, OAT-FM cannot significantly improve straightness (NPE) further. However, thanks to the relaxation mentioned above, OAT-FM achieves significantly better data fitting (lower $\mathcal{W}_2$ distance), as it can fine-tune the flow with greater flexibility to better match the data distribution.
>
> In summary, we hope the above response resolves your concerns and makes you confident in raising your score. Thanks in advance for your consideration.

---

> ### Comment · Reviewer_FKVq · 2025-11-27
> **Thank you for the answer**
>
> I thank the authors for the detailed response, which addresses the majority of my concerns and questions. Sitll, I am not absolutely confident about raising my score.
> In eq. (9) you make a simplification which significantly affects the OAT theory. I think, it should be stated clearer in the main text, to avoid wrong impression. Also, I think, an ablation study should be conducted (e.g., which extends your ablation in Table 4, or, at least, on low-dim OT benchmark, section 4.1), which studies the effect of considering "honest" cost in eq. 9, to demonstrate that indeed the training/performance become less stable.
> Overall, I have a positive impression of the work, and plan to raise my score if my suggestion (which I think to be important) is properly taken into account.

---

> > ### Author Response · Authors · 2025-11-28
> > **Response to Reviewer FKVq's remaining concern for Eq.9**
> >
> > We sincerely thank you very much for the detailed and positive feedback.
> >
> > Your rigorous approach to Eq. 9 and its derivation motivates us to double-check our code. We find that **our implementation does consider the full cost** $12 \Vert x_1 - x_0 - \frac{v_{\theta} (x_1,1) + v_{\theta} (x_0,0)}{2} \Vert_2^2 + \Vert v_{\theta}(x_1,1) - v_{\theta}(x_0,0) \Vert_2^2$, **without ignoring any term**. (See the source code in our supplementary file, **`oat_distance.py'**, **Lines 54-63**). We pasted the pseudo code below for a quick review.
> >
> > ```python
> > # 1. Impulse term: ||v1 - v0||^2
> > impulse_terms = torch.cdist(v0, v1, p=2.0).pow(2)
> >
> > # 2. Curvature term: 12 * || B - A ||^2
> > A_src = x0 + 0.5 * v0
> > B_tgt = x1 - 0.5 * v1
> > curvature_terms = 12.0 * (torch.cdist(A_src, B_tgt, p=2.0).pow(2))
> >
> > # 3. Full OAT cost
> > oat_dist_sq = curvature_terms + impulse_terms
> > ```
> >
> > We have revised Eq.(9) (**Lines 254-260**) and the corresponding derivations (**Appendix C.1**, **Lines 949-966**), ensuring they match our code implementation.
> >
> > In addition, to resolve your concern completely, we compared the OAT-FM using the **simplified cost** ($\Vert x_1-x_0\Vert_2^2+(x_1-x_0)^{\top}(v_{\theta}(x_1,1)+v_{\theta}(x_0,0))$) with our real implementation using the above **full cost** on CIFAR-10 experiment. The results below show that these two costs lead to similar performance. The reason for this interesting phenomenon is left as our future work.
> >
> > |Method|NFE (Dopri5)|FID|NFE (Euler)|FID|
> > |:---|:---:|:---:|:---:|:---:|
> > |FM|147|3.71|100|4.60|
> > |+OAT simplified|136|**3.51**|100|3.93|
> > |+OAT full|135|3.54|100|**3.92**|
> > |I-CFM|149|3.67|100|4.40|
> > |+OAT simplified|137|**3.48**|100|3.79|
> > |+OAT full|138|**3.48**|100|**3.78**|
> > |OT-CFM|132|3.64|100|4.49|
> > |+OAT simplified|128|3.49|100|3.74|
> > |+OAT full|126|**3.46**|100|**3.73**|
> >
> > We apologize for this careless mistake, but at the same time, we should emphasize that 1) this mistake does not affect the correctness of our experimental results and the superiority of our method, and 2) even considering the full cost, the complexity of our method is still $O(B^2 \log B)$ when batch size is $B$, which achieves the acceleration of OAT indeed.
> >
> > We hope this clarification addresses your concern about the OAT cost and helps you feel confident about raising your score.

---

### Official Review · Reviewer_yNNm · 2025-11-01

**Soundness:** 3
**Presentation:** 3
**Contribution:** 3
**Rating:** 6
**Confidence:** 3

**Summary:**

### **Summary of Contribution**

This paper introduces OAT-FM, a novel method for improving existing Flow Matching (FM) generative models. The core idea is to re-frame the flow-straightening objective through the lens of Optimal Acceleration Transport (OAT), which operates in the product space of samples and velocities ($\mathcal{X} \times \mathcal{V}$) rather than just the sample space ($\mathcal{X}$).

The authors argue that standard Optimal Transport (OT) based FM methods enforce a constant-velocity flow ($a=0$), which is a *sufficient* but not *necessary* condition for a straight trajectory. OAT-FM instead aims to satisfy a necessary and sufficient condition (Proposition 1: acceleration parallel to velocity) by minimizing the total acceleration.

The method is proposed as a "two-phase" paradigm:

1.  **Phase 1:** Train any standard FM or diffusion model to obtain "relatively reliable" velocity estimates.
2.  **Phase 2:** Fine-tune this model using the OAT-FM objective. This objective is a bi-level optimization problem where:
    * The **lower-level** computes an optimal coupling in the product space. Critically, this is shown to be decomposable and simplifies to a standard OT problem with a modified cost function (Eq. 9) that incorporates the endpoint velocities ($v_0, v_1$) from the Phase 1 model.
    * The **upper-level** minimizes an acceleration-based loss ($\mathcal{L}_{OAT}$) over this new coupling.

Experiments demonstrate that this Phase 2 fine-tuning consistently improves the performance (e.g., FID) of strong baselines (including FM, EDM, and the SOTA SiT-XL) with a very small number of additional training steps.

**Strengths:**

1.  **Strong Theoretical Motivation:** The paper provides a clear and compelling theoretical motivation. The distinction between the *sufficient* (constant velocity) and the *necessary and sufficient* (acceleration parallel to velocity) conditions for straight flows is a sharp insight. Grounding the method in the physics of OAT and the Vlasov equation (as noted in Table 1) provides a solid theoretical foundation that is distinct from prior art.

2.  **Pragmatic and Novel Formulation:** The two-phase paradigm is a very clever solution to the "chicken-and-egg" problem. The authors admit OAT-FM is "fragile" from scratch (a key limitation) but deftly turn this into a strength by proposing it as a fine-tuning method that *leverages* the reliable velocity estimates from a pre-trained model.

3.  **Computationally Tractable:** The primary technical hurdle—solving an OT problem in the high-dimensional product space $\mathcal{X} \times \mathcal{V}$—is elegantly overcome. By using the pre-trained model $v_\theta$ to deterministically define $v_0$ and $v_1$, the authors successfully decompose the coupling and reduce the lower-level problem to a tractable OT problem in $\mathcal{X}$ with a modified cost (Eq. 9). This makes the method computationally feasible.

4.  **Strong Empirical Validation:**
    * **SOTA Improvement:** The method demonstrates consistent, measurable improvements on top of very strong baselines. Improving EDM on CIFAR-10 (FID 1.96 $\rightarrow$ 1.93) and SiT-XL on ImageNet (FID 2.11 $\rightarrow$ 2.05) with minimal fine-tuning (12K batches and 5 epochs, respectively) is a significant and practical achievement.
    * **Excellent Ablation Study:** Table 4 provides a strong defense of the method's design. It clearly shows that *both* the new $\mathcal A_2^2$ based coupling (lower-level) and the $\mathcal{L}_{OAT}$ objective (upper-level) are essential. The finding that the standard $\mathcal W_2^2$ coupling *destroys* the pre-trained EDM model (FID 1.96 $\rightarrow$ 8.77) is a powerful piece of evidence for the authors' hypothesis.

**Weaknesses:**

1.  **Limited Generality:** The method's reliance on a "relatively reliable" pre-trained model is its primary weakness. It is an *improver*, not a standalone generative modeling algorithm. The paper is transparent about this, but it fundamentally limits the scope of the contribution.

2.  **Scalability Bottleneck:** The method inherits the $\mathcal{O}(B^2)$ mini-batch coupling complexity from OT-CFM. This is a known scalability bottleneck that is not solved here. As models and batch sizes (B) grow, this quadratic cost becomes a significant practical barrier. This is mentioned as future work but remains a present weakness.

3.  **Unexplored Sensitivity:** The entire method's success hinges on the quality of the Phase 1 model's velocity estimates. The paper does not investigate the sensitivity to this quality. What happens if the Phase 1 model is mediocre? Is there a performance threshold below which OAT-FM fine-tuning provides no benefit, or even leads to degradation? This is a crucial practical question that is left unanswered.

4.  **Marginal Gains on Large-Scale Tasks:** While the improvements on SiT-XL are consistent, they are marginal (e.g., FID 2.11 $\rightarrow$ 2.05). A skeptic could argue this is a minor tweak for a large computational cost (solving a mini-batch OT problem at every step). The visual improvements in Figures 4-6 are extremely subtle, reinforcing this point.

**Questions:**

1.  **On Sensitivity to Phase 1 Quality:** Could you expand on the "fragility" of training OAT-FM from scratch? More importantly, can you quantify the relationship between the quality of the Phase 1 model and the efficacy of the Phase 2 fine-tuning? For example, if you take FM models from various checkpoints (e.g., 100K, 200K, 400K batches), does OAT-FM provide more, less, or equal improvement? This would help users understand when to "apply" OAT-FM.

2.  **Analysis of Coupling Failure:** The result in Table 4 where the $\mathcal{W}_2^2$ (standard OT) coupling catastrophically degrades the EDM model is fascinating. What is your hypothesis for why this happens? Does the standard OT plan create pairs $(x_0, x_1)$ that are so "unnatural" for the model's learned velocity field that it breaks the dynamics? Does the OAT-FM coupling (Eq. 9), by incorporating $\bar{v}$, act as a regularizer that keeps the couplings "closer" to the model's existing expectations?

3.  **Practical Cost vs. Benefit:** For the ImageNet experiments, what was the practical wall-clock training time for the 5 epochs of OAT-FM fine-tuning? How does this compare to the time for 5 epochs of *continued* standard training on the original SiT-XL objective? This is essential for evaluating if the marginal FID gain is worth the computational cost of the mini-batch OT.

4.  **Hyperparameter** $\alpha$**:** What value of the hyperparameter $\alpha$ (from Eq. 6) was used in the experiments? Was it fixed to $\alpha=2/3$ as motivated by Theorem 3, or was it tuned as a hyperparameter? How sensitive are the results to this choice?

---

> ### Author Response · Authors · 2025-11-22
> **Response to Reviewer yNNm (Part 1)**
>
> Thank you for your appreciation of our work. We have revised our paper based on your comments, adding more explanations and experiments. Below, we answer your questions one by one.
>
> **Q1: Generality of two-phase training.**
>
> **A1:** The two-phase nature of OAT-FM is a strategic design choice that enhances practical utility rather than limiting its contribution, aligning with the field's most effective paradigms.
>
> 1. **The dominance and necessity of two-phase paradigms.** In the current landscape of generative modeling, two-phase methods are valuable, not a weakness. In practice, many two-phase methods, e.g., model distillation and Reflow, are dominant in this field [1,2].
>
> 2. **In methodology, OAT-FM is a new paradigm of two-phase methods, enforcing a necessary and sufficient condition for flow straightness in the OAT framework.** Unlike ReFlow, OAT-FM is trained on original data, eliminating the risk of distribution drift and improving performance with minimal training overhead.
>
> 3. **Robustness to phase-1 model.** As shown in the following A3, OAT-FM performs well even when the phase-1 model is not well-trained, demonstrating its robustness. Moreover, this result shows its potential for training from scratch, which we leave as future work.
>
> [1] Flow Straight and Fast: Learning to Generate and Transfer Data with Rectified Flow, 2023
>
> [2] Consistency Models, 2023
>
>
> **Q2: Scalability of OAT-FM**
>
> **A2:** Minibatch-OT indeed suffers from the scalability issue when the batch size is very large. This problem is common for all primal-formed OT algorithms, not specific to our OAT-FM. Below, we would like to emphasize the contributions of our work in computation.
>
> 1) **Make OAT from inapplicable to practical.** Given a batch of $B$ samples and $B$ velocities, the original complexity of OAT is at least $\mathcal{O}(B^3)$, which is too high to practical applications. We leverage a decomposable formulation (Eq.(9)) to reduce the complexity to $\mathcal{O}(B^2\log B)$ (equivalent to that of standard OT), making an intractable method practical.
>
> 2) **Demonstrate the empirical efficiency of OAT-based FM.** Most existing OT-based FM methods are trained and tested on simple tasks (e.g., CIFAR-10). In our work, we scale OAT-FM to ImageNet-level image generation. To further resolve your concern, we provide detailed training costs of OAT-FM in the following table (**Table 10 in the revised paper**), demonstrating that the quadratic scaling of the mini-batch OT solver introduces modest overhead in the context of FM.
>
> - **Competitive Speed & Negligible Memory:** OAT-FM maintains 73%-80% speed of the baseline (SiT-XL) while keeping memory consumption almost identical. In practice, a global batch size of 1024 or 2048 is sufficient for ImageNet and most generation tasks, which our method handles effortlessly.
>
> - **Short Total Training Time:** Although OAT-FM introduces additional computations per step, it is highly data-efficient, requiring only 5 epochs for the phase-2 training, whose total training duration is $<$ 4 hours on 8 A100 GPUs.
>
> ||GPU|Batchsize|Peak Allocated/Reserved Memory (GB)|Speed (samples/s)|Time (Hours per 5 epochs)|
> |-|-|-|-|-|-|
> |SiT-XL|A6000×8|128×8|35.57/44.61| 276 |6.1|
> |+OAT-FM|A6000×8|128×8| 35.61/44.62|225|7.5|
> |SiT-XL|A100×8|128×8|35.57/44.49|584|2.9|
> |+OAT-FM|A100×8|128×8|35.61/44.49|440|3.8|
> |SiT-XL|A100×8|256×8|55.67/72.61|614|2.7|
> |+OAT-FM|A100×8|256×8|55.91/72.83|452|3.7|
>
> To further improve per-step computation speed, we are actively investigating the dual formulation of OAT, as Optimal Flow Matching (OFM) does. We leave this as our future work.
>
> **Q3: Robustness to phase-1 model**
>
> **A3:** OAT-FM demonstrates robust performance gains across varying levels of pre-trained model quality.
>
> 1. **Consistent improvement across convergence stages.** As detailed in our revised sensitivity analysis (**Table 9 in Appendix D.3.2**, pasted below for a quick review), we applied OAT-FM to models (FM, I-CFM, OT-CFM) trained for varying durations (100K to 400K batches). OAT-FM consistently reduces FID and NFE across all stages. Notably, it benefits even under-converged models, confirming that a near-perfect velocity field is not a prerequisite.
>
> 2. **Significant training efficiency.** OAT-FM accelerates convergence significantly. For instance, an I-CFM model trained for 300K batches refined with OAT-FM (only 1K batches) achieves an FID of 3.44, outperforming the stronger, fully converged OT-CFM baseline (400K batches, FID 3.64).
>
> ||100K NFE|100K FID|200K NFE|200K FID|300K NFE|300K FID|400K NFE|400K FID|
> |-|-|-|-|-|-|-|-|-|
> |FM|140|6.11|140|4.26|143|3.88|147|3.71|
> |+OAT-FM (1K)|**135**|**5.60**|**132**|**3.96**|**134**|**3.63**|**135**|**3.54**|
> |I-CFM|140|5.97|140|4.13|140|3.81|149|3.67|
> |+OAT-FM (1K)|**131**|**5.60**|**137**|**3.95**|**134**|**3.44**|**138**|**3.48**|
> |OT-CFM|138|6.23|133|4.40|134|3.93|132|3.64|
> |+OAT-FM (1K)|**128**|**6.02**|**126**|**4.18**|**128**|**3.71**|**126**|**3.46**|

---

> ### Author Response · Authors · 2025-11-22
> **Response to Reviewer yNNm (Part 2)**
>
> **Q4: The significance of improvement in large-scale tasks:**
>
> **A4:** We respectively disagree with the comment that our improvement is not significant in high-dimensional tasks. In particular, as shown in Figure 3, when cfg=4.0, a common setting for generating high-resolution images, our method achieves significant improvements in FID and sFID (gains of around 1). **Table 11 in Appendix D.4.2** provides a more detailed numerical comparison. In addition, **for the CIFAR-10 experimental results in Table 3, the improvement from FM to FM+OAT-FM (3.71->3.54) is more significant than that from FM to OT-CFM (3.71->3.64).**
>
> To further resolve your concern, in Table 9 of the revised paper, we evaluate the efficacy of OAT-FM fine-tuning (applied to only 1K batches) on FM, I-CFM, and OT-CFM checkpoints trained for varying durations (100K to 400K batches). The results have also been pasted in **A3** for quick review, demonstrating that our method consistently achieves significant improvements, even when the initial models are not well-trained.
>
> In Figure 4, in the revised paper, we present enlarged visual results for SiT-XL and SiT-XL+OAT-FM for better comparison and analyze their differences (**Lines 512-521**). In particular, when starting from the same noise, in some cases (e.g., the "Bottle" and "Parrot" shown in Figure 4), applying OAT-FM improves image details, maintains the main semantic and spatial content generated by SiT-XL, and suppresses hallucination. In the other cases (e.g., "Lion" and "Car" shown in Figure 4), applying OAT-FM results in the model generating entirely different images compared to those produced by SiT-XL. This phenomenon indicates that the coupling associated with OAT-FM differs from that of the original model, altering the flow trajectories from noise to data in two distinct ways. For flow trajectories whose endpoints follow the data distribution well, OAT-FM simply straightens them while maintaining their endpoints. For flow trajectories whose endpoints are undesired (e.g., in the "Lion" generated by SiT-XL), OAT-FM significantly changes their directions, leading to new endpoints that may better fit the data distribution.
>
> **Q5: The reason for the failure of classic OT when fine-tuning EDM**
>
> **A5:** As we discussed in the revised paper (**Lines 454-465**), our OAT-based coupling improves EDM performance, whereas the classic OT-based coupling results in catastrophic performance degradation. This phenomenon reveals the essential difference between OT and OAT in the velocity smoothness. In particular, EDM learns a score function under the assumption that the noise at time $t$ is independent and isotropic Gaussian noise. When reformulating EDM as an FM model (See **Appendix D.1**), this assumption means that the velocity field is smoothed [1]. The OT-based coupling, however, achieves static, non-Markovian transport between noise and data in the sample space, without any smoothness constraint on the velocity. In contrast, the OAT-based coupling matches the "sample-velocity" tuples at $t=0$ with those at $t=1$ in the product space $\mathcal{X}\times\mathcal{V}$, solving the dynamic-form OAT problem in Eq.(4) equivalently. The objective of Eq.(4) reveals the isotropic Gaussian prior of the acceleration, which leads to a smoothed velocity [2]. Consequently, our OAT-based coupling follows the smoothness assumption and is suitable for refining EDM.
>
> [1] Score-Based Generative Modeling through Stochastic Differential Equations, 2021.
>
> [2] Second-order models for optimal transport and cubic splines on the Wasserstein space, 2019.
>
> **Q6: Practical Cost vs. Benefit**
>
> **A6:** As shown in **A2**, applying OAT-FM merely introduces mild additional computation costs. At the same time, the significance of improvement has been verified in **A4**.
>
> **Q7. The setting of hyperparameter $\alpha$**
>
> **A7:** The parameter $\alpha$ balances velocity alignment against the acceleration penalty, as shown in Lines 211-214. Empirically, we observe robust performance within the range $\alpha \in [0.7, 0.8]$. In our experiments, we set $\alpha=0.7$ for toy tasks, $0.75$ for CIFAR-10, and $0.80$ for ImageNet. In the revised paper, we have added the settings in the Appendix.
>
> Note that setting $\alpha=12/13$ in Eq.(6) aims to connect our loss to the cost of OAT in Eq.(5), which emphasizes the motivation of our loss design. In Theorem 3, setting $\alpha=2/3$ leads to an OAT-based lower bound of our loss, which helps verify the rationality of our loss. These theoretical values do not necessarily correspond to the optimal settings for stable and effective training in practice.
>
> In summary, we hope the above response resolves your concerns and enhances your confidence in raising your score. Thanks in advance for your consideration.

---

> ### Comment · Reviewer_yNNm · 2025-11-27
>
> Thanks for the detailed response and most of my concerns have been addressed. It's a solid work and definitely meets the standard of ICLR. I'm happy to keep my score and raise the confidence, but I cannot find the button to edit the review. I will do it once the bug is fixed.

---

> ### Author Response · Authors · 2025-12-03
> **Thank You for Your Feedback**
>
> Thank you for your support and for raising the confidence score. We appreciate your recognition that our work meets ICLR's standards, as well as your time and effort in the review process.

---

### Author Response · Authors · 2025-11-26
**Gentle Reminder**

Dear Reviewers,

We sincerely appreciate your thoughtful and constructive feedback. We have carefully addressed all concerns in our point-by-point responses and have uploaded a revised version of the paper, with new content highlighted in blue.

As the end of the author-reviewer discussion phase is approaching, we kindly hope you have had a chance to review our updates. If there are any remaining questions or concerns, we are happy to discuss further and provide additional clarification.

Best regards,

Authors of ICLR 2026 Submission 10172

---

### Author Response · Authors · 2025-12-03
**Summary of the Rebuttal**

Dear Reviewers, ACs, SACs, and PCs,

Over the past few days, we have done our best to address reviewers' concerns and revise our submission, including adding suggested analytical experiments, clarifying the rationale for the proposed method, and providing a more detailed analysis of model performance.

Although our work received mixed scores (**6**, **6**, **4**, and **4**) in the initial review, **all four reviewers appreciated the novelty of our method and its theoretical and experimental contributions.** Their concerns are listed as follows.

- **Reviewer yNNm (Score: 6)**

1. The generality of two-phase training.

2. The reason for the failure of classic OT when fine-tuning EDM.

3. The scalability and efficiency of the method.

4. The hyperparameter selection.

- **Reviewer FKVq (Score: 4)**

1. The theoretical motivations and connections of our formulations (Eqs. 4, 6, and 7).

2. The validity of the acceleration strategy (Eq. 9).

3. The comparison of flow straightness.

- **Reviewer byYL (Score: 4)**

1. The theoretical solidness of OAT and its connection to our method.

2. The refinement behavior during fine-tuning.

3. The significance of improvements in high-dimensional generation tasks.

4. The comparison with OFM.

- **Reviewer BrGa (Score: 6)**

1. The scalability and efficiency of the method.

2. The hyperparameter selection.

3. The specific strategy for target network updates.


**Summary of Our Response**

During the discussion phase, we have addressed the reviewers' questions point-by-point and supplemented our work with additional experiments. Key updates include:

1. **Clarifying the theoretical foundations.** We detailed the motivation for our loss design (Eq. 6) and its connection to the OAT problem (Eq. 7) in **Lines 227-231**. We clarified the solidity of the OAT theory itself (**Line 215**) and the correctness of our acceleration strategy (Eq. 9), specifically elucidating why OAT succeeds in fine-tuning EDM where classic OT does not (**Lines 455-465**).

2. **Validating the stability of the two-phase paradigm.** In **Appendix D.3.2**, we added experiments that track the refinement behavior of OAT-FM as the number of fine-tuning iterations increases. The results confirm that the improvement achieved by our refinement is consistent and stable.

3. **Demonstrating robustness across training durations.** In **Appendix D.3.2**, we applied OAT-FM to Phase-1 models (FM, I-CFM, OT-CFM) trained for varying durations (100K to 400K batches) on CIFAR-10. OAT-FM consistently reduces FID and NFE across all stages, proving its robustness to the pre-trained model's quality.

4. **Verifying scalability and computational efficiency.** In **Table 10**, we clarified how our decomposable formulation (Eq. 9) makes the OAT problem practical. Detailed cost analyses demonstrate the efficiency of our implementation in the context of Flow Matching.

5. **Benchmarking against OFM.** In **Table 7**, we compared OAT-FM with OFM on low-dimensional benchmarks in a one-step generation setting, demonstrating the advantage of our method.


**Summary of Reviewer Feedback**

We received feedback from three reviewers, and **we are pleased to highlight that all engaged reviewers expressed a positive attitude toward our work:**

1. **Reviewers yNNm and BrGa are satisfied with our response and clearly claim that they maintain their positive scores.** Reviewer yNNm further confirmed that our work meets the ICLR standard, and he/she planned to raise the confidence score from 3 to 4.

2. **Reviewer FKVq stated that our response addressed most of his/her concerns and expressed a positive impression of the work.** He/she planned to raise the score pending further clarification on the OAT cost and Eq. (9). In our follow-up, we have revised Eq.(9) (Lines 254-260), the corresponding derivations (Appendix C.1, Lines 949-966), and added more experiments, ensuring our derivation matches our code implementation. We are confident that these revisions have comprehensively addressed this final concern. Due to the security incident, the reviewer is unable to provide final feedback or update their score. We kindly ask the AC to consider the positive trajectory of this review and the reviewer's explicit promise to raise the score upon this clarification.

3. **Reviewer byYL acknowledged the promising direction and novelty of our work in the initial review. Unfortunately, he/she did not provide any feedback to our rebuttal.** Therefore, we rely on AC's judgment to confirm that our response fully addresses the reviewer's concerns.

In summary, we believe we have effectively addressed the reviewers' concerns. In light of this, we respectfully hope that our detailed rebuttals will be favorably considered in the following phase. We are grateful for the time and effort you and the reviewers have dedicated, and we thank you in advance for your fair and careful assessment.

Best regards,

Authors of Submission 10172

---

### Meta-Review · Area_Chair_5bGd · 2026-01-08

**Summary:**

This paper introduces OAT-FM, a two-phase refinement framework for flow and diffusion models inspired by Optimal Acceleration Transport (OAT). The key insight is that minimizing acceleration in the joint sample–velocity space provides a necessary and sufficient condition for straight trajectories, improving upon OT-based flow matching which enforces constant velocity as a sufficient condition. Because reliable endpoint velocities are not available early in training, the method is explicitly designed as a fine-tuning procedure applied to pretrained models.

The authors provide a practical formulation that reduces the OAT coupling in the product space to a standard minibatch OT problem, making the method computationally feasible. Experiments on 2D benchmarks, CIFAR-10, and ImageNet show consistent improvements over strong baselines (including EDM and SiT-XL) with relatively modest additional training.

**Reviewer Concerns:**

The rebuttal successfully clarifies several practical and empirical issues raised during the review. In particular, the authors explain how the original OAT problem in the joint sample–velocity space is reduced to a standard minibatch OT problem with the same asymptotic complexity as OT-CFM, and provide additional runtime and memory analyses demonstrating that the fine-tuning overhead is moderate even at ImageNet scale. Added experiments show that the two-phase refinement is stable over training iterations and that performance improvements are consistent across pretrained models of varying quality, alleviating concerns about brittleness. Hyperparameter choices and target network update strategies are also clarified, improving reproducibility and reducing ambiguity in the implementation.

Despite these improvements, the overall scope and impact of the contribution remain limited. OAT-FM is explicitly a refinement method that depends on already strong pretrained models, which constrains its generality and positions it as an incremental improvement rather than a standalone modeling advance. Empirically, although gains are consistent, they are relatively modest on large-scale benchmarks, raising questions about cost–benefit tradeoffs given the added complexity of minibatch OT. Finally, the inherited quadratic scaling of minibatch OT remains a practical limitation that is not resolved in this work.

**Reviewer Scores:**

Reviewer yNNm and Reviewer BrGa are likely to keep their positive scores. Reviewer byYL not sure. Reviewer FKVq is likely to raise the score.

---

### Decision · Program_Chairs · 2026-01-26

Reject